# Unexpected similarities between C9ORF72 and sporadic forms of ALS/FTD suggest a common disease mechanism

Erin G Conlon[1], Delphine Fagegaltier[2], Phaedra Agius[3], Julia Davis-Porada[1], James Gregory[2], Isabel Hubbard[2], Kristy Kang[2], Duyang Kim[2], The New York Genome Center ALS Consortium, Hemali Phatnani[2], Neil A Shneider[4], James L Manley[1]*

[1]Department of Biological Sciences, Columbia University, New York, United States; [2]Center for Genomics of Neurodegenerative Disease, New York Genome Center, New York, United States; [3]New York Genome Center, New York, United States; [4]Department of Neurology, Columbia University Medical Center, New York, United States

**Abstract** Amyotrophic lateral sclerosis (ALS) and frontotemporal dementia (FTD) represent two ends of a disease spectrum with shared clinical, genetic and pathological features. These include near ubiquitous pathological inclusions of the RNA-binding protein (RBP) TDP-43, and often the presence of a GGGGCC expansion in the *C9ORF72* (C9) gene. Previously, we reported that the sequestration of hnRNP H altered the splicing of target transcripts in C9ALS patients (Conlon et al., 2016). Here, we show that this signature also occurs in half of 50 postmortem sporadic, non-C9 ALS/FTD brains. Furthermore, and equally surprisingly, these 'like-C9' brains also contained correspondingly high amounts of insoluble TDP-43, as well as several other disease-related RBPs, and this correlates with widespread global splicing defects. Finally, we show that the like-C9 sporadic patients, like actual C9ALS patients, were much more likely to have developed FTD. We propose that these unexpected links between C9 and sporadic ALS/FTD define a common mechanism in this disease spectrum.

DOI: https://doi.org/10.7554/eLife.37754.001

*For correspondence:
jlm2@columbia.edu

Group author details:
The New York Genome Center ALS Consortium See page 21

## Introduction

Amyotrophic lateral sclerosis (ALS) is a fatal neurodegenerative disease that afflicts approximately 1 in 400 people worldwide (*Alonso et al., 2009*). Diagnosis is predicated on symptoms of motor neuron (MN) loss from the primary motor cortex (upper MNs), brainstem and spinal cord (lower MNs), which results in progressive weakness and loss of motor control, and eventually paralysis and death (*Rowland and Shneider, 2001*). While predominantly a motor disorder, many ALS patients experience cognitive and behavioral changes, and ~10–15% of ALS patients meet formal criteria for a diagnosis of frontotemporal dementia (FTD; *Lattante et al., 2015*; *Ling et al., 2013*). FTD is a heterogeneous condition marked by the degeneration of neurons in the frontal cortex, and patients exhibit a range of cognitive, behavioral and language deficits (*Seelaar et al., 2011*).

The co-occurrence of ALS and FTD is genetically determined in some cases, most often by the dominantly inherited *C9ORF72* (C9) hexanucleotide expansion (*DeJesus-Hernandez et al., 2011*; *Renton et al., 2011*). The presence of a single copy of expanded C9 increases sharply the chance that an ALS-afflicted individual will be diagnosed with FTD, from 10 to ~50% (*Byrne et al., 2012*). While consensus on precisely how the expanded sequence acts as a pathological unit has remained elusive, we have shown that the repeat-containing RNA can directly sequester the RNA-binding

protein (RBP) hnRNP H, leading to dysregulation of pre-mRNA splicing (*Conlon et al., 2016*). Generally, our and others' evidence for proteinaceous aggregates, seeded by RNAs and which are prone to undergo phase transitions (*Fay et al., 2017*; *Jain and Vale, 2017*; *Schwartz et al., 2013*), may offer a bridge from aggregation - as a classical observation of neurodegenerative disease (*Ginsberg et al., 1998*; *Ross and Poirier, 2004*) - to a biophysical mechanism at the origins of ALS/FTD, in cases both with and without disease-associated mutations (*Alberti and Hyman, 2016*; *Conlon and Manley, 2017*; *Mackenzie et al., 2017*; *Murakami et al., 2015*).

Histopathologic similarities also link ALS and FTD on this disease spectrum. In almost all ALS and over half of FTD patients, the distributions of pathological cytoplasmic accumulation of the hnRNP-like DNA/RNA binding protein TDP-43 tend to correlate with areas of neurodegeneration (*Baloh, 2011*). Although the observed aggregates of TDP-43 come in several forms (*Scotter et al., 2015*), translocation and nuclear clearance events are of particular interest, as they have suggested a loss-of-function (LOF) mechanism (*Vanden Broeck et al., 2014*), predicted to impact mRNA processing and transport (*Humphrey et al., 2017*; *Ling et al., 2015*; *Polymenidou et al., 2011*; *Tollervey et al., 2011*). While TDP-43 aggregation could reflect the downstream effects of cell death rather than the cause, the existence of ALS-causing mutations in the gene that encodes TDP-43 (*TARDBP*) suggests that the protein has a deterministic role in the ALS/FTD spectrum (*Scotter et al., 2015*). Between the extremes of bystander and instigator, it is possible that changes to TDP-43 reflect an important indicator of upstream processes that influence RBPs more pervasively.

In all ALS/FTD cases with TDP-43 pathology, aggregation may be driven by any number of factors that impact the protein's ability to form stoichiometrically correct interactions with itself or with other nuclear proteins and RNAs. For example, disease-linked changes affecting the solubility properties of other RBPs (*Harrison and Shorter, 2017*), including hnRNP H (*Conlon et al., 2016*), have the potential to affect solubility of RBPs they interact with, like TDP-43 (*Freibaum et al., 2010*; *Ling et al., 2010*), although whether this occurs in ALS/FTD patient brains is unknown. Extending this idea, downstream changes to the transcriptome, particularly splicing patterns, may result from combinatorial deficiencies of several RBPs (*Huelga et al., 2012*; *Mohagheghi et al., 2016*), making it difficult to pinpoint the initiating event or distinguish specific effects of TDP-43 LOF.

In this study, we have investigated whether the biochemical 'signature' we described previously for C9ALS might extend to a broader spectrum of ALS/FTD patients. For this, we analyzed splicing and RBP solubility in 50 postmortem human brains from patients with sporadic ALS, FTD with motor neuron disease (MND) and ALS-FTD (collectively, sALS/FTD), all of which were negative for known disease mutations, including C9. Unexpectedly, we identified splicing dysregulation of known hnRNP H targets (*Conlon et al., 2016*) in a large fraction of patients, and used the extent of splicing dysregulation to divide the patients into two groups, like-control and like-C9. Notably, we found that the like-C9 patient brains also had higher amounts of insoluble hnRNP H, the amount of which directly correlated with the magnitude of splicing dysregulation. Importantly, several other abundant RBPs linked to ALS, including TDP-43, showed the same patterns of insolubility as hnRNP H. Consistent with this, we found pervasive splicing dysregulation genome wide in like-C9 brains. Intriguingly, the like-control cases with lowest hnRNP H/TDP-43 insolubility displayed almost no splicing dysregulation, despite the ubiquitous presence of histopathological TDP-43 inclusions. Finally, and importantly, we show that like-C9 patients, similar to actual C9ALS patients, were much more prone to develop dementia than were like-control patients. Together, our data define unanticipated biochemical similarities between a large fraction of sALS/FTD cases and C9ALS/FTD, suggesting a common mechanism underlying disease pathogenesis.

## Results

### Patient stratification based on graded inclusion of hnRNP H-regulated exons

C9 expansion is the most common known cause of ALS/FTD. However, a large majority of cases, especially amongst the 90% that are sporadic in origin, are of unknown etiology. When initially

characterizing hnRNP H-dependent splicing events, we compared C9ALS/FTD patients to neurological controls and SOD1 ALS (not believed to proceed through a mechanism of RBP dysfunction or aggregation; *Ling et al., 2013*), but not to sALS/FTD (*Conlon et al., 2016*). Indeed, sALS/FTD is a mechanistic unknown and may constitute a wide range of molecular subtypes. Given that TDP-43 and hnRNP H appear to interact (*Appocher et al., 2017*; *Ling et al., 2010*; *Sephton et al., 2011*) in a manner that is largely RNA dependent (*Freibaum et al., 2010*), it is reasonable to speculate that changes in RNA binding and/or solubility of one might impact the other, leading to more complex patterns of splicing dysregulation.

We first set out to determine whether splicing patterns specific to C9ALS/FTD could be distinguished from those related to ALS/FTD more generally. To do this, we initially analyzed whether sALS/FTD patients display splicing changes similar to those we described in C9 patients (*Conlon et al., 2016*) (*Figure 1a*). Specifically, we measured alternative exon inclusion by $^{32}$P-RT-PCR of oligo(dT)-selected RNA samples purified from post-mortem cerebellum from 50 sporadic patients: 32 classified as ALS (without cognitive/behavioral involvement), seven as ALS-FTD (symptoms of both disorders during the patient's life), and 11 as FTD with motor neuron disease (FTD-MND) defined by a lack of clinical ALS symptoms yet the presence of pathological MN degeneration at autopsy. We also included two patients carrying pathogenic SOD1 variants, and four patients carrying *C9* repeat expansions, for a total of 56 cases (*Supplementary file 1A*).

To assay hnRNP H-dependent splicing changes, we selected 18 known hnRNP H target exons that we previously found are differentially included in C9ALS/FTD compared to individuals without neurological symptoms and SOD1 ALS controls (*Figure 1a*). Four of the targets were hnRNP H-repressed exons, meaning hnRNP H normally represses inclusion when physiological levels of the protein are high, and the other 14 were hnRNP H-enhanced exons, which respond with decreased inclusion when hnRNP H levels are low (*Figure 1b*). Results obtained with two enhanced exons (from *ARRB2* and *PPP1R12C*) and one repressed exon (from *OS9*) are shown in *Figure 1c–e*. Representative gels for all other genes analyzed are shown in *Figure 1—figure supplement 1*. The data revealed dramatic differences across many of the patient samples. We observed several patients that consistently had the lowest levels of enhanced exon inclusion for all 14 exons, and noticed that these same patients had the highest levels of repressed exon inclusion (e.g. boxed FTD-MND samples 9–13; *Figure 1c–e*). Other patients consistently displayed the opposite trend, or a milder version of both.

We next wished to classify the patients as either having or not having hnRNP H-dependent splicing changes. For this, we first averaged the percent inclusion of all the enhanced exons to define a single composite splicing score (CSS). We then compared the CSS values of sporadics to those of a set of neurological normal and SOD1-ALS controls, and to C9ALS/FTD patients. Interestingly, we observed that for these same 14 exons, the controls had a very close range of values, while the C9 + range was centered significantly lower and with a wider distribution (*Figure 1f*). Only a single control point deviated significantly from the mean; this point corresponds to the second SOD1-ALS patient and is likely due to poor RNA quality. Due to the rarity of SOD1-ALS post-mortem samples, we decided against discarding this patient. This is the only sample in the entire set where RNA quality was compromised.

We then wished to determine which, if any, of the sporadics had exon inclusion values in the range only accessible to C9+, and not control, brains. However, the wide range of CSS values of the C9+ group in theory made it impossible to classify a patient as C9+ or control based on a CSS similar to a control value. The converse, however, was not true: control samples had a close range of observed values, and all the patients with significantly lowered values were disease afflicted. Thus, we defined 'like-control' as any patient with a CSS within three standard deviations (SD) of the non-ALS mean, since this range should encompass 99.7% of values within a normally distributed population. 'Like-C9' was then defined as all the patients with CSS values lower than this value. These two categories split the 50 patients into two groups of roughly equal size (*Figure 1f*). Comparison of like-control (n = 24) and like-C9 (n = 26) revealed a significant difference (p<0.05) between these two categories for each of the 18 genes, with only one exception, *RPL10* (p=0.09) (*Figure 1g*).

We next wished to determine, for every individual hnRNP H-regulated exon we analyzed, if each patient displayed similar relative change in inclusion values as for all the other exons. To this end, we first re-calculated the CSS for each combination of 13 enhanced exons and plotted a best-fit line for each gene in the case where it was excluded from the average, using each patient's percent exon

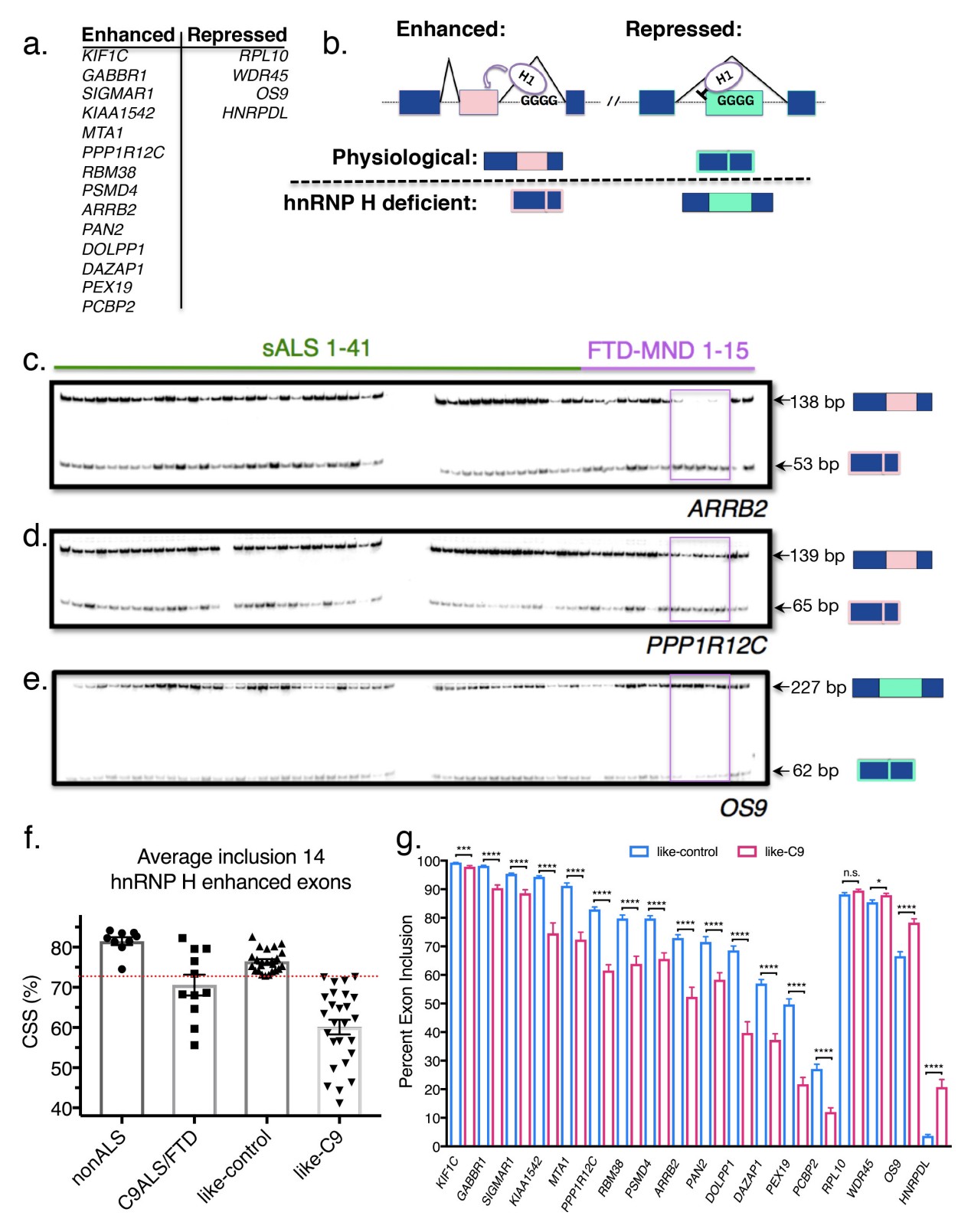

**Figure 1.** Patient stratification into like-control and like-C9 groups by aberrant splicing of known hnRNP H-target exons. (a) List of hnRNP H target exons, separated into enhanced and repressed categories. (b) (Top) Simplified diagram of enhanced and repressed exons with relative binding of hnRNP H (H1). (Bottom) Depiction of expected product when H1 levels are at normal physiological levels (above dashed line) or low (hnRNP H deficient; below dashed line). (c–e) Representative PCR gels of all patient samples for two enhanced exons (ARRB2 and PPP1R12C) and one repressed

*Figure 1 continued on next page*

*Figure 1 continued*

exon (*OS9*). Pink boxes show samples F09-F13 that consistently exhibited large magnitudes of change in inclusion. (**f**) Graph of CSS by group: nonALS, C9ALS, like-control and like-C9. Each point represents the CSS of an individual patient. Red dashed line marks the cutoff of 3 standard deviations below the control mean. Error bars are plotted to the SEM. (**g**) Comparison of percent exon inclusion of 18 hnRNP H-target exons between like-control and like-C9 patient replicate values. Exons are ordered by enhanced exons (*KIF1C-PCBP2*) followed by repressed exons (*RPL10-HNRPDL*). Error bars are plotted to the SEM. (t test p value: *=0.05, **=<0.01, ***=<0.0001, ****=<0.0001).

DOI: https://doi.org/10.7554/eLife.37754.002

The following figure supplement is available for figure 1:

**Figure supplement 1.** RT-PCR of hnRNP H target genes.

DOI: https://doi.org/10.7554/eLife.37754.003

inclusion value for that gene as a Y-coordinate. Indeed, we found that each individual gene followed the trend of the patients' average CSS (*Figure 2a*; simplified to show three best- and one worst-fit lines, all others are shown in *Figure 2—figure supplement 1*), with the same patients consistently showing the most severe changes, and others consistently showing the most mild. Some genes had extremely good fit to a linear model (*GABBR1*, r-square = 0.90, *ARRB2*, r-square = 0.88), while others displayed the trend without as good a fit (*PAN2*, r-square = 0.40). We also averaged the four hnRNP H repressed exons, and plotted this value as a Y coordinate (*Figure 2b*). As expected, the patients with the lowest values of hnRNP H-enhanced exon CSS had the highest levels of average repressed exon inclusion, as the data fit well to an inverse linear relationship (r-square = 0.75).

## Like-C9 sALS/FTD display increased hnRNP H insolubility

The above results defined a cohort of patients that displayed graded splicing dysregulation similar to what we observed in C9ALS patients. We next asked whether these patients also displayed increased levels of sarkosyl-insoluble hnRNP H, indicative of functional sequestration (see *Conlon et al., 2016*). To this end, we performed biochemical fractionation of motor cortex from all cases where tissue was available, with two independent replicates per case (*Figure 2c*). While we performed RNA analysis with cerebellum for abundance and RNA quality reasons, for fractionation we used motor cortex as it is a primary site of disease-specific degeneration. Given the historical range of our cohort, for some samples motor cortex was unavailable, and thus we omitted those cases from our analysis. We then analyzed equivalent amounts of each fraction, soluble (SOL), sarkosyl soluble (SS), and sarkosyl insoluble (SI), by Western blot (WB), and calculated the percentage of insoluble hnRNP H in each case. Using 39 samples (18 like-control and 21 like-C9), we found that the average percent insoluble hnRNP H was 69.2 in like-C9s, and 38.9 in like-control, or 1.78 times higher in like-C9, a highly significant increase in aggregated protein (p<0.0001) (*Figure 2d*; all average percent insolubility values are shown by patient in *Supplementary file 1A*). This enrichment closely resembles the 1.9-fold increase we previously reported in seven C9ALS/FTD (56.1% insoluble) compared to controls (30.0% insoluble).

We next wished to quantify the relationship, if any, between splicing severity and hnRNP H insolubility. We therefore performed the same regression analysis as we did above with splicing changes, except with percent sarkosyl-insoluble hnRNP H. We found that the general trend was for patients with large changes in splicing to have high insoluble hnRNP H (*Figure 2e*; r-square = 0.58). These observations suggest that hnRNP H is differentially sequestered, with patients showing a gradient of decreasing soluble hnRNP H that correlates with increasing splicing dysfunction. The correspondence between hnRNP H's biochemical solubility and its predicted transcriptomic effects across distal brain regions from the same patients strongly supports the notion that this is an intrinsic, brain-wide property of each patient. Reinforcing this view, we also examined available postmortem pathological reports from 26 of the sporadic ALS/FTD patients. Significantly, 19 of these were noted to exhibit focal loss of Purkinje neurons in the cerebellum (*Supplementary file 1A*), confirming that regions of the brain undergoing robustly measured splicing defects did indeed experience neuronal degeneration. This further justifies our choice to correlate changes measured in the cerebellum with changes in the cortex.

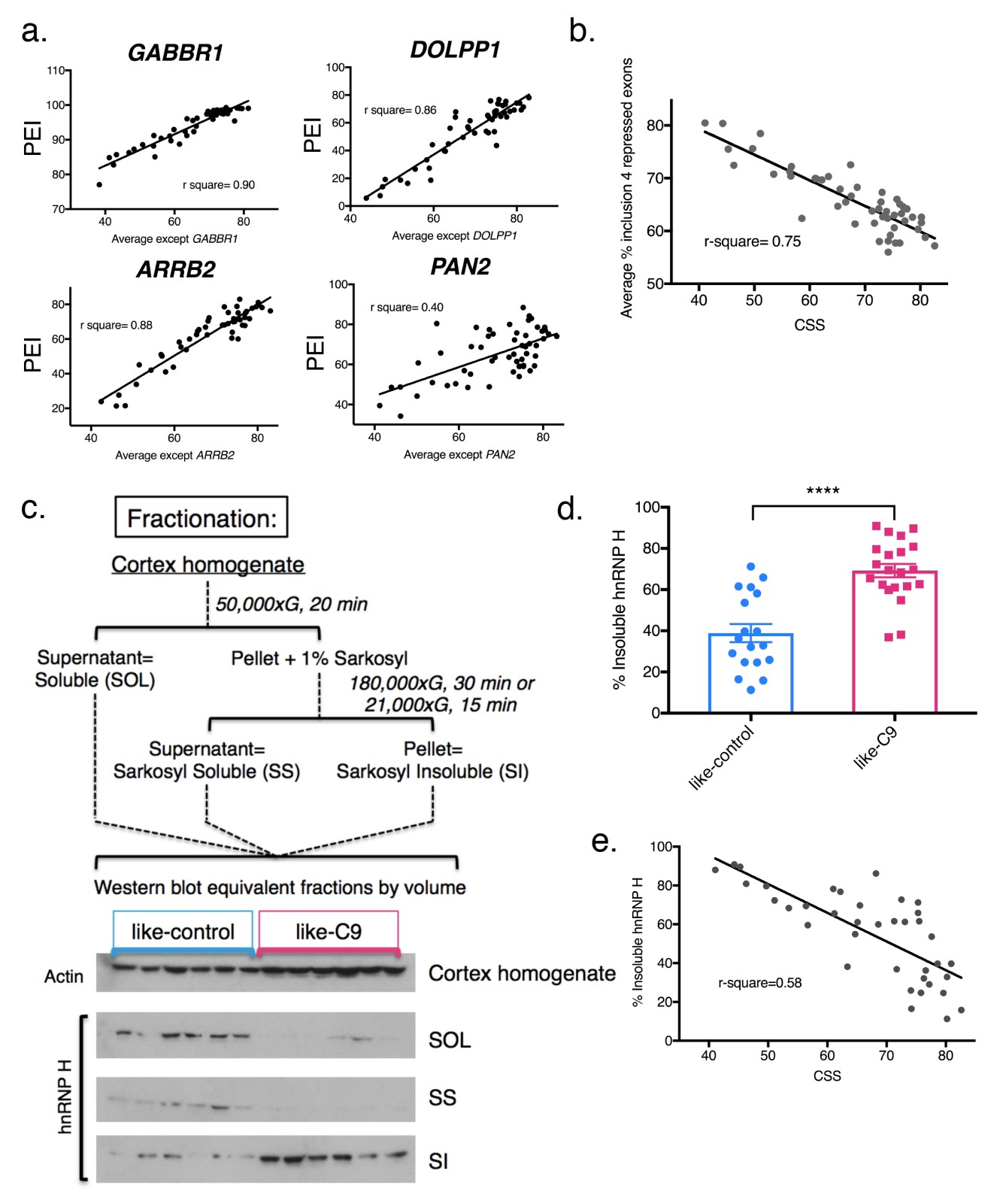

**Figure 2.** Differential concentration of hnRNP H due to insolubility explains graded splicing dysregulation in patient brains. (**a**) Linear regression of percent exon inclusion (PEI) for each enhanced target exon compared to the average inclusion of all other enhanced exons. Clockwise from upper left: *GABBR1*, *DOLPP1*, *PAN2*, *ARRB2*, with r-square = 0.90, 0.86, 0.40, 0.88, respectively. (**b**) Linear regression of percent exon inclusion for the average of four repressed target exons compared to the average inclusion of all enhanced exons. R-square = 0.75. (**c**) (Upper) Diagram of biochemical

*Figure 2 continued on next page*

*Figure 2 continued*

fractionation and resultant fractions. (Lower) Representative example of 6 like-control and six like-C9 samples, western blotted for actin (cortex homogenate fraction) and hnRNP H (SOL, SS, SI). (d) Summary graph percent insoluble hnRNP H in motor cortex of like-controls versus like-C9s. Samples for which motor cortex was unavailable were excluded from this analysis. Each point represents a single patient with replicate values. Error bars are plotted to the SEM. (t test p value: *=0.05, **=<0.01, ***=<0.0001, ****=<0.0001). (e) Linear regression of percent insoluble hnRNP H compared to CSS, r-square = 0.58.
DOI: https://doi.org/10.7554/eLife.37754.004

The following figure supplement is available for figure 2:

**Figure supplement 1.** Linear regression of hnRNP H enhanced exons.
DOI: https://doi.org/10.7554/eLife.37754.005

## Insolubility of hnRNP H correlates with that of the ALS-linked RBPs TDP-43 and FUS

We previously provided evidence that the insolubility of hnRNP H in C9ALS brains was due to formation of aggregated assemblies with long, repetitive, G quadruplex-forming RNA (*Conlon et al., 2016*). However, in the sporadic cases analyzed here, the source of hnRNP H insolubility/aggregation is unknown. To investigate this question, and to understand the significance of the observed gradient of hnRNP H insolubility and splicing dysregulation, we considered the possibility that it is related to insolubilities of other proteins with similar functions and/or domain structures. For example, as mentioned above, TDP-43 displays histopathological evidence of cytoplasmic accumulation/aggregation in virtually all ALS cases. Is this relevant to the biochemical insolubility we observed with hnRNP H?

To measure insolubilities of more protein targets, including TDP-43, we selected a subset of 20 patients: the 10 most like-control and the 10 most like-C9, based on highest and lowest CSS, respectively (and omitting cases where motor cortex was unavailable for fractionation) (*Figure 3a*). First, to demonstrate clearly the gradient pattern of these samples for hnRNP H insolubility, we performed simultaneous WB of all 20 samples and each of the three fractions (soluble, sarkosyl soluble and sarkosyl insoluble) (*Figure 3b*). We also performed splicing analysis of a new hnRNP H-regulated target, *ACHE* (*Nazim et al., 2017*), further illustrating the relationship between hnRNP H insolubility and splicing dysregulation (*Figure 3c*).

Having established these 20 cases as representative of the hnRNP H gradient, we used the motor cortex fractions derived from them for WB for three other ALS/FTD-implicated hnRNPs: TDP-43, FUS (also known as hnRNP P2), and hnRNP A1 (*Figure 3d*). These proteins all belong to a common yet dynamic interactome (*Markmiller et al., 2018*); FUS has been reported to interact with hnRNP H (*Reber et al., 2016*) and hnRNP A1 (*Kim et al., 2013*), while hnRNP A1 has been reported to interact cooperatively with hnRNP H to modulate splice site selection (*Fisette et al., 2010*). Likewise, TDP-43 was shown to interact with the structurally related hnRNP A1 to repress exon inclusion (*Buratti et al., 2005*) and has also been suggested to self-associate (*Polymenidou et al., 2011*; *Tollervey et al., 2011*) to influence a set of targets it shares with FUS (*Lagier-Tourenne et al., 2012*). As a control, we also blotted for GAPDH, which seemed unlikely to be insoluble.

Strikingly, for each RBP target we found significant increases in insoluble protein in the like-C9s, matched by significant decreases in soluble protein (*Figure 3e*). Indeed, no insoluble FUS could be detected in any like-control patient, while it was present in all of the like-C9s (between 6.1 and 69.1%). Intriguingly, we found that the correlations between CSS and percentage insoluble protein for hnRNP H, TDP-43, FUS and hnRNP A1 each fit remarkably well to a line (r-square = 0.89, 0.79. 80 and 0.90, respectively) (*Figure 3f*). Analysis of GAPDH revealed no insoluble protein in any of the 20 cases (*Figure 3e*) confirming that insolubility of these RBPs was not due to an artifact of fractionation or global protein insolubility in like-C9s.

The above results were unanticipated, and we therefore wished to examine properties of the insoluble fractions further. One possibility was that the high force (180,000 x G) used to generate the sarkosyl insoluble fractions might have resulted in an artificial enrichment of RBPs in the pellet of some patients, and thus, the similar insolubilities of these targets may not reflect physiological interactions. To examine this, we decreased the sedimentation force we used to isolate the aggregated proteins, by separating the sarkosyl insoluble pellet at 21,000 x G for 15 min instead of 180,000 x G

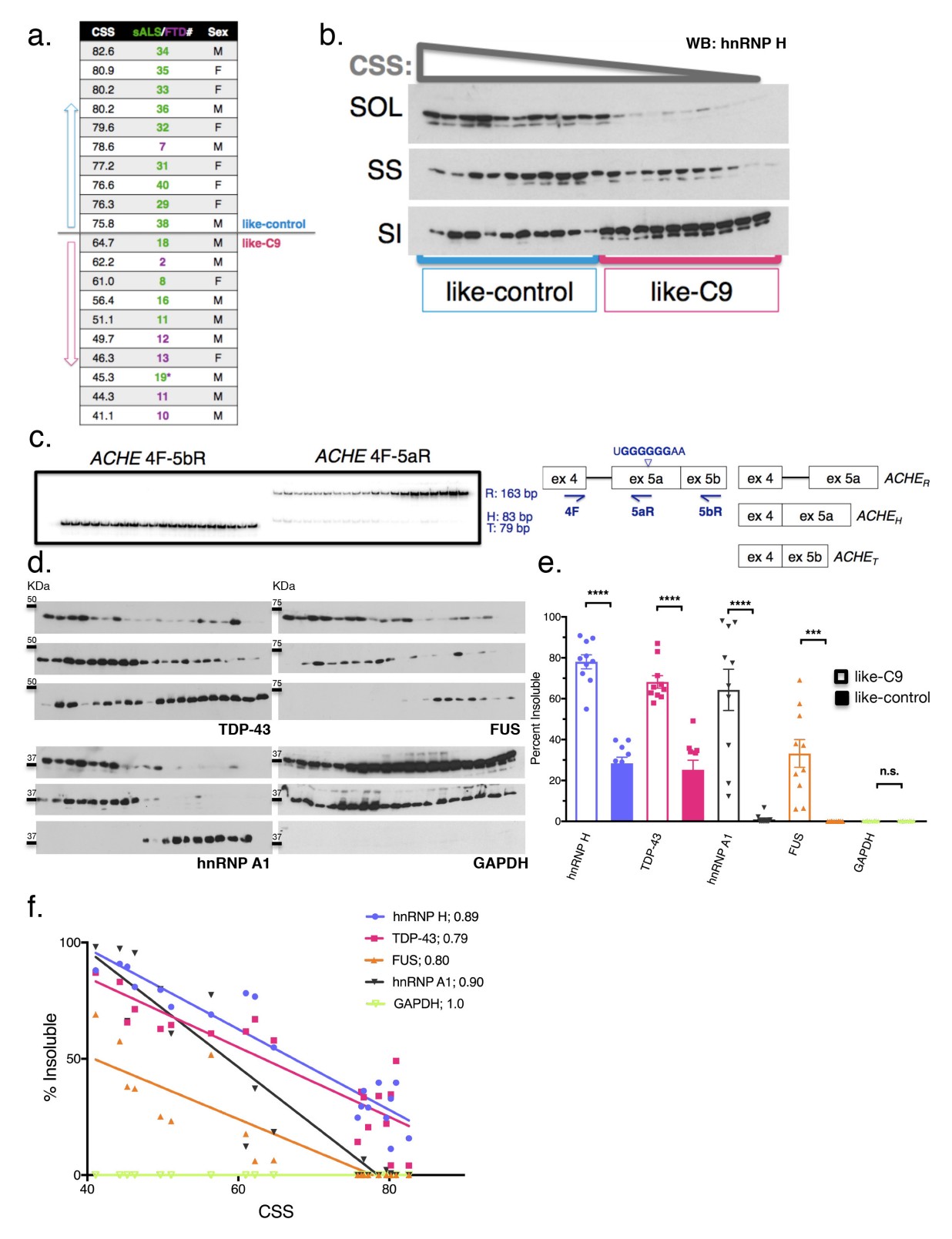

**Figure 3.** Insolubility of hnRNP H correlates with insolubility of additional RBPs. (a) CSS scores and identifiers for 10 most like-control and 10 most like-C9 patients for which motor cortex was available. These patients were loaded in this descending order for all parts of this figure. (b) Fractionation (180,000 x G) of 20 sALS-FTD patients blotted for hnRNP H. SOL = soluble, SS = sarkosyl soluble and SI = sarkosyl insoluble. (c) Left: gel of ACHE alternative splicing in 20 sALS-FTD patients, loaded in order of highest to lowest CSS. PCR products are identified at right. Right: diagram of primers

*Figure 3 continued on next page*

*Figure 3 continued*

used and hnRNP H binding motif. (**d**) Fractionation shown in (**b**) with western blotting for the following targets (clockwise from upper-left): TDP-43, FUS, GAPDH, hnRNP A1. Each target is shown with three panels representing SOL, SS and SI, from top to bottom. (**e**) Quantification of percent insoluble protein (180,000 x G) in these 20 cases, with replicate values. Error bars are plotted to the SEM. (t test p value: *=0.05, **=<0.01, ***=<0.0001, ****=<0.0001). (**f**) Linear regression of percent insoluble hnRNP H, TDP-43, FUS, hnRNP A1 and GAPDH for 20 sALS-FTD patients, plotted against CSS. R-square is listed in legend.

DOI: https://doi.org/10.7554/eLife.37754.006

for 30 min. We performed simultaneous fractionation and WB of the 20 cases described above, and expanded our targets beyond hnRNP H, TDP-43, hnRNP A1 and FUS to include three additional proteins that we would not expect *a priori* to be insoluble: Histone H3, C9ORF72 itself and splicing factor U2AF65. We also included one more target protein with broad relevance to MN disease, SMN, which has been reported to interact with RBPs in the context of RNA transport granules (*Hua and Zhou, 2004*; *Thomas et al., 2011*; *Murakami et al., 2015*).

The results of this analysis confirm and extend the observations we made with the 180,000 x G fractionation. They support the idea that the insolubility we initially described for hnRNP H (*Figure 4a*) extends to other MN disease-related proteins (*Figure 4b–e*), but is not a general property of like-C9 brains. The non-disease associated proteins (*Figure 4f and g*) did not display any insolubility, except for U2AF65 (*Figure 4h*), which had low levels of insoluble protein in six of ten like-C9s, or 9.2% on average (*Figure 4i*). Intriguingly, a fraction of SMN was insoluble in all the like-C9 cases with a range from 8.1–61.8%, or 31.1% on average, but in none of the like-controls. For comparison, hnRNP H and hnRNP A1 were insoluble in like-C9s with a range from 16.0 to 77.5% and 29.4 to 94.7% (*Figure 4i*). Despite the important role of SMN in snRNP assembly, and thus in splicing per se, its relatively low average insolubility in like-C9s suggests it is only partially deficient and thus present at levels sufficient to facilitate snRNP assembly, and far higher than levels seen in SMA (*Lefebvre et al., 1997*). The low levels of U2AF65 and presence of the four abundant hnRNP proteins in the insoluble fraction is supportive of the like-C9s having alterations in the soluble concentration of a limited set of auxiliary splicing factors, yet maintaining sufficient concentrations of the core factors necessary for splicing.

## Reciprocal insolubilities of hnRNP H and TDP-43 reveal TDP-43 LOF is exclusive to like-C9s

The similarity between hnRNP H and TDP-43 insolubilities in the like-C9 samples suggests that a common process or factor may be functioning to bring about their aggregation to correspondingly severe degrees. Supporting this idea, insolubility of each protein displayed an inverse relationship with CSS (*Figure 4j*), and correlation analysis of the measured insolubilities of all these RBPs (at both 180,000 and 21,000 x G) amongst this set of 20 patients revealed highly significant correlation for each pairwise combination (*Figure 4k*). The top pairwise combinations between different proteins were between hnRNP H and TDP-43 at both 180,000 and 21,000 x G (*Figure 4k*; Spearman r = 0.95 for both). Supporting the idea that primary insolubility of hnRNP H can drive insolubility of these other factors, we performed fractionation at 21,000 x G with 13 C9 patients and found similar correlation between hnRNP H and TDP-43 (Spearman r = 0.71) (*Figure 4—figure supplement 1*).

One explanation for the similar degrees of hnRNP H and TDP-43 insolubility in the sporadic and C9 patient brains is that they coexist in the same physical aggregates. To test this hypothesis, we performed co-immunoprecipitation (co-IP) of hnRNP H from eluates of the sarkosyl insoluble pellets of two like-C9 brains using a TDP-43 antibody for IP (see Materials and methods). Indeed, TDP-43 antibody successfully co-IPed hnRNP H, while a negative control antibody (FLAG-M2) or beads alone did not (*Figure 4l*). These results support a reciprocal relationship between insolubility of hnRNP H and TDP-43, whereby aggregates initially seeded by one protein can recruit the other, and vice versa.

The possibility that a reciprocal relationship exists between hnRNP H and TDP-43 prompted us to consider that insoluble TDP-43 is upstream of hnRNP H insolubility in like-C9s cases. Etiologically, this is appealing, since hallmark TDP-43 histopathology is presumed to reflect underlying biochemical insolubility, and aggregation of TDP-43 into this non-functional and/or mislocalized state is often given as a rationale for mechanistic studies that have sought evidence for TDP-43 LOF

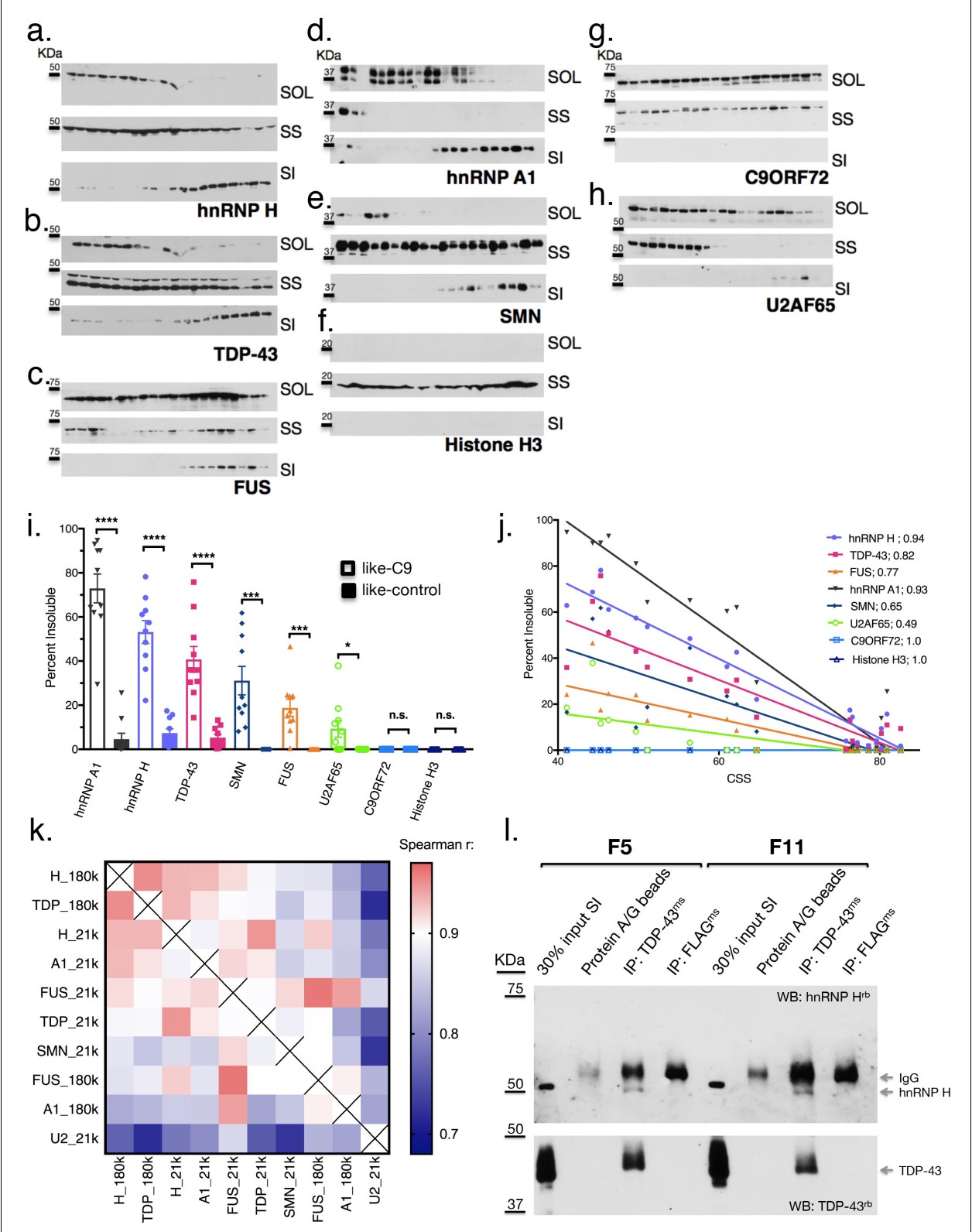

**Figure 4.** Like-control ALS/FTD patients have undetectable levels of insoluble RBPs after low-speed fractionation. (a–h) Low-speed (21,000 x G) fractionation experiment with 20 patients listed in *Figure 3a*. Western blot hnRNP H (a), TDP-43 (b), FUS (c), hnRNP A1 (d), SMN (e), Histone H3 (f), C9ORF72 (g), U2AF65 (h). SOL = soluble, SS = sarkosyl soluble, SI = sarkosyl insoluble. (i) Quantification of low-speed centrifugation of 20 patients with replicate values. Error bars are plotted to the SEM. (j) Linear regression of percent insoluble protein for targets in (a–h) plotted against CSS. R-square is

*Figure 4 continued on next page*

*Figure 4 continued*

listed. (**k**) Heatmap of spearman correlation coefficients for each pairwise comparison of percent insoluble protein at 180,000 or 21,000 x G (180 k and21k, respectively). (**l**) Western blots of hnRNP H (top) and TDP-43 (bottom) following IP with protein A/G magnetic beads alone, TDP-43 antibody, or control (FLAG) antibody from two patient brains (F5 and F11). The abbreviations ms and rb stand for mouse antibody and rabbit antibody, respectively. Protein species are indicated by arrows at right, and positions of size markers are shown on the left.

DOI: https://doi.org/10.7554/eLife.37754.007

The following figure supplements are available for figure 4:

**Figure supplement 1.** Insoluble hnRNP H and TDP-43 correlate in C9 patient brains.

DOI: https://doi.org/10.7554/eLife.37754.008

**Figure supplement 2.** Quanitification of TDP-43 splicing targets.

DOI: https://doi.org/10.7554/eLife.37754.009

(*Polymenidou et al., 2011*; *Lagier-Tourenne et al., 2012*; *Ling et al., 2015*). However, our data present a significant contradiction of this tenet, as 21,000 x G fractionation revealed TDP-43 to be, on average, only 5.3% insoluble in the 10 like-control patients. Meanwhile, 10 patients that were on the same clinical and pathological spectrum yet classified as like-C9 had on average almost eight times as much insoluble TDP-43 (40.8 versus 5.3%, t test p<0.0001) (*Figure 4j*). Fractionation of the same samples at 180,000 x G revealed a majority of TDP-43 to be insoluble in like-C9s (68.2%) compared to a minority in like-controls (25.2%; p<0.0001; *Figure 3e*).

The above differential TDP-43 insolubility is in sharp contrast with the much more uniform histopathology. Of the 23 sporadic cases for which we possessed data for pathological analysis by immunohistochemistry with TDP-43 antibodies, only two were found to lack detectable TDP-43 inclusions in motor regions (*Supplementary file 1A*). One of these, s30, was a like-control case, while the other, F10, was the most like-C9 case (lowest CSS), and had an average of 61.8% insoluble TDP-43 based upon two measurements at 180,000 and 21,000 x G each (*Figures 3f* and *4i*, respectively). Conversely, case 36 had only 2.3% insoluble TDP-43 based upon two measurements at 180,000 and 21,000 x G each (*Figures 3f* and *4i*, respectively), yet TDP-43 pathology was indeed evident (*Supplementary file 1A*).

The above observation suggests that insolubility of TDP-43, as well as hnRNP H, is not related to TDP-43 histopathology. Given that hnRNP H insolubility is correlated with splicing dysfunction of a set of validated targets, we next asked whether TDP-43 LOF-associated splicing changes also occurred more in the like-C9 cases than in like-controls. To assay TDP-43-dependent splicing, we selected targets from previously published TDP-43 knockdown (KD) and CLIP studies (*Tollervey et al., 2011*) and analyzed them as above by $^{32}$P RT-PCR (*Figure 4—figure supplement 2*). For 18 targets that produced appropriate products, we compared the percent inclusion values from the six most like-control and six most like-C9 patients (six highest and lowest CSS, respectively). Using a t-test for each event, and a more permissive cutoff for statistical significance (p<0.10), we found that 10 of the 18 events displayed significant changes. Of these events, eight were as predicted consistent with the like-C9 group having lower concentrations of functional TDP-43, while the other two displayed the opposite trend. These results suggest that TDP-43-regulated splicing events demonstrate differential exon inclusion amongst sALS/FTD patients with TDP-43 pathology, with the patients demonstrating biochemically insoluble TDP-43 displaying greater TDP-43 LOF.

## RNA-sequencing reveals widespread splicing defects that correlate with protein insolubility

Our investigation into splicing dysfunction initially focused specifically on hnRNP H splicing targets. However, the results described thus far imply that the splicing changes due to decreased levels of functional hnRNP H are complemented by shortages of other RBPs such as TDP-43. We therefore analyzed transcriptome-wide splicing on a larger cohort of patients (28 sALS/FTD cases described above for which hnRNP H solubility and clinical data was available and an additional 13 confirmed C9ALS/FTD cases) using RNA-seq of RNA purified from cerebellum. As controls, we used a combined group of six neurologically normal individuals and five SOD1-ALS individuals, with only two of these 11 samples coming from our previously categorized control group (*Supplementary file 1A*). The decision to include SOD1 in the control group was made because SOD1-ALS represents the

only known form of ALS marked by the absence of TDP-43 (or FUS) inclusions (*Mackenzie et al., 2007*).

To analyze splicing patterns globally and without bias, we regrouped the patients in a manner independent of previously measured hnRNP H splicing. Instead, we divided the 28 sALS/FTD patients into three groups based on clinical diagnosis and hnRNP H insolubility: ALS-hnRNP H low insolubility (ALSlow; 7 patients), ALS-hnRNP H high insolubility (ALShigh; 13 patients) and FTD (FTD; 8 patients) (*Supplementary file 1A*). The first two groups consisted of patients with ALS only, and with measured hnRNP H insolubility being below or above 50%, respectively (*Figure 5a*). The FTD group consisted of those patients with clinical diagnoses of FTD. Interestingly, this group contained the five most severely like-C9 patients. Of the 11 FTD-MND patients in our cohort that were subjected to RNA-seq, only two, F3 and F7, had insolubility below 50%, and therefore were not included in this analysis, but rather used as test cases to ascertain the predictive power of the clustering with respect to insolubility. A fourth patient group consisted of C9ALS and ALS/FTD patients (C9). The average percentage insoluble hnRNP H (180,000 x G) in each group was 26.4 in ALSlow, 59.4 in C9, 67.3 in ALShigh and 79.6 in FTD (*Figure 5b*). In order to rule out differential hnRNP H levels in the cerebellum of these patients, we performed WB of tissue from 26 of the samples, and observed that there were no changes in overall hnRNP H abundance between these groups (*Figure 5c*).

We next compared differential splicing (DS) events in each of the four patient groups to the controls. For this analysis, we used the annotation-free method Leafcutter (*Li et al., 2018*). Variations in intron splicing were considered significant in ALS/FTD when the Percentage Spliced Index (PSI) compared to control differed by more than 10% ($|\Delta PSI| \geq 0.1$) with a 10% FDR cutoff (see Materials and methods). To confirm our assumption that neurological normal and SOD1 ALS could be grouped as one, we compared these two groups and indeed found very few (23) detectable splicing differences ($|\Delta PSI| \geq 0.1$). The paucity of splicing defects in these patients contrasted with 1300–1400 DS events in ALShigh and C9 groups and several thousand in the FTD group (*Figure 5d*; events listed in *Supplementary file 1B*), with 488 events overlapping between C9, FTD and ALShigh. Comparison of events without filtering for the magnitude of the change in PSI revealed an even greater proportion of overlapping events between the FTD, C9 and ALShigh groups (*Figure 5—figure supplement 1*). As validation of our like-C9 nomenclature, with or without filtering respectively, only 27 or 8% of all events detected in C9 were unique to that group; the remainder were detected in ALShigh, FTD or both (like-C9 collectively). Of the remaining 73% of events that were detected in C9 (with filtering), approximately half (48%) were shared with both other groups, while 40 and 12% of events, respectively, were exclusively shared with FTD or ALShigh (*Figure 5d*). The higher degree of overlap of C9 with FTD likely reflects the fact that with few exceptions, this group had the largest average change in PSI for each event. Furthermore, the ALSlow group had an insignificant number of events, even without filtering, upholding our definition of like-control for splicing in general.

We next wished to visualize the degree of change in splicing of all events by patient. For this, we produced heatmaps of the normalized difference in PSI for all the splicing events that differed significantly between control and any other group ($|\Delta PSI| \geq 0.1$), excluding the very large number of events that differed exclusively between control and FTD (*Figure 5d*). Patients were organized according to the above groupings, and within each group they were further arranged by increasing hnRNP H insolubility from left to right (*Figure 5e*, gray-scale) to visualize the relationship between hnRNP H insolubility and changes in splicing. To extend our analysis, we also included 23 additional samples from C9, ALS and FTD patients that were not included in one of the groups above (*Supplementary file 1A*), creating an additional group of ALS patients (ALS) not categorized as ALSlow or ALShigh.

Hierarchical clustering of splice junctions revealed two distinct groups of events that displayed opposite directions of normalized PSI change in the controls versus the FTDs (*Figure 5e*). As expected, the ALSlow group was found to be most similar to controls, and the FTD group the most different, reflecting the extreme ends of the insolubility spectrum. Within the C9 and ALShigh groups, the patients displayed a gradient of patterns from low to high insolubility that generally mirrored the pattern of PSI from control to FTD.

We next sought to identify significant splicing changes in our dataset that best correlate with hnRNP H insolubility. We therefore ranked events according to their correlation to insolubility

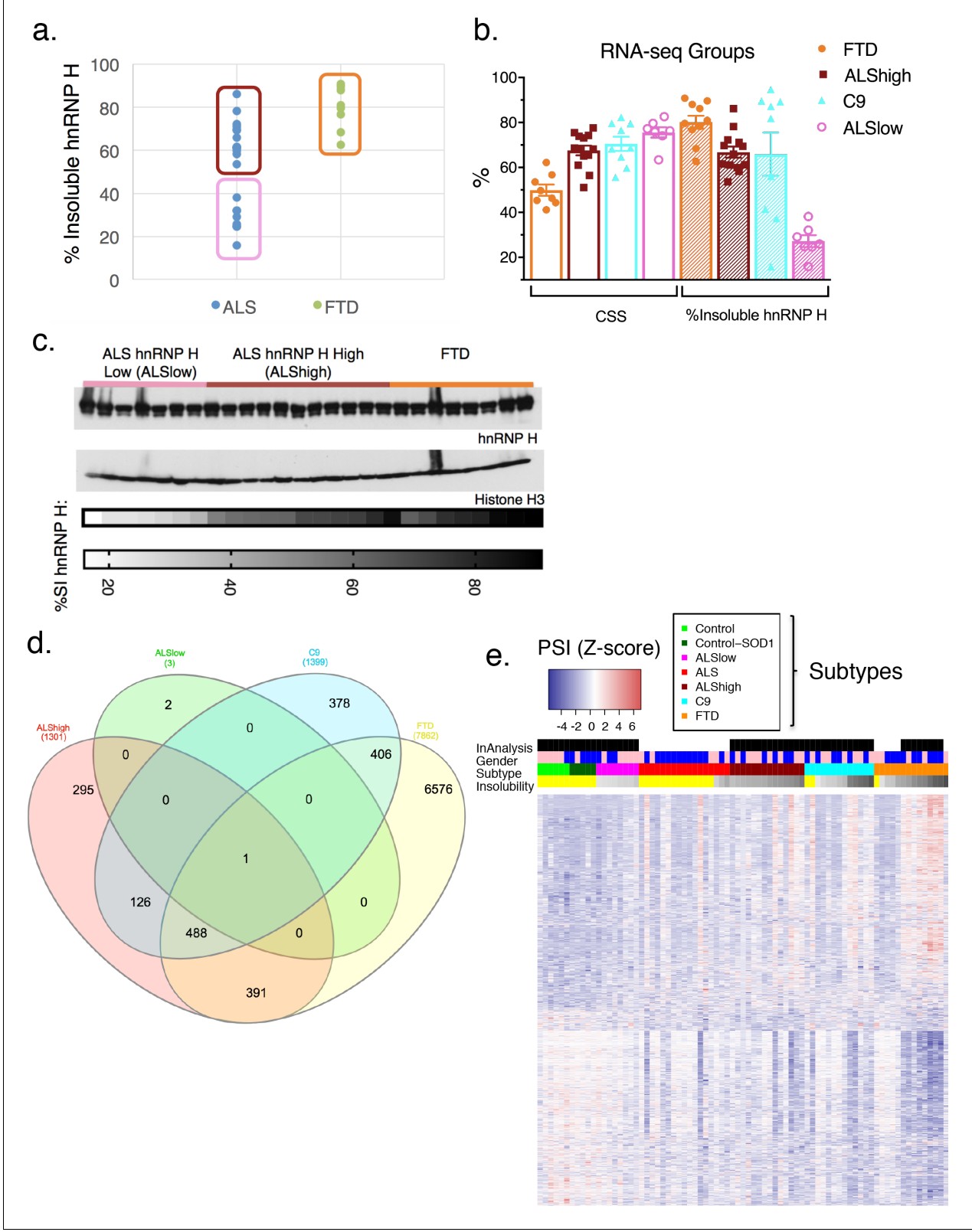

**Figure 5.** hnRNP H insolubility defines patient groupings for RNA-sequencing analysis. (a) Percent insoluble hnRNP H of 28 samples in RNA-seq analysis reveals 50% as a natural boundary between ALSlow (7 patients) and ALShigh (13 samples) samples. Percent insoluble hnRNP H of FTD patients (8 patients) is also shown. (b) RNA-seq groups plotted by CSS and percent insoluble hnRNP H. All samples used in RNA-seq differential splicing (DS) analysis groups are shown. (c) Western blots of hnRNP H (top) and histone H3 (middle) from whole-tissue homogenates from cerebellum of 26/28

*Figure 5 continued on next page*

*Figure 5 continued*

samples used in RNA-seq analysis. Two ALShigh cases (s1 and s20) were excluded in order to fit all samples on one gel. Grayscale heatmap displays percent insoluble hnRNP H (bottom). (**d**) Venn diagram of all splicing events found to be significantly different between each patient RNA-seq group and the combined control/SOD1ALS group with PSI magnitude filtering (|ΔPSI| ≥ 0.1). (**e**) Heatmap of all events in (**d**), except for the 6576 events found to have significant DS exclusively in FTD. Blue-red color scale represents Z-score for normalized PSI. Four rows above heatmap display whether each sample was included in the groupings used for the identification of DS events, termed 'In analysis' (black = yes, white = no), gender (pink = female, blue = male), Patient subtype (see key), and percent insolubility hnRNP H (180,000 x G), ranging from light (low insolubility) to dark (high insolubility) gray. Yellow is shown for cases without insolubility data. Patients are arranged by subtype, with increasing insolubility from left to right, where data is available.

DOI: https://doi.org/10.7554/eLife.37754.010

The following figure supplement is available for figure 5:

**Figure supplement 1.** DS events without filtering.

DOI: https://doi.org/10.7554/eLife.37754.011

(*Supplementary file 1C*; 100 most correlated events) and plotted several. The results (*Figure 6a*) demonstrate that splicing of these transcripts was, as expected, affected by variations in soluble hnRNP H levels. Amongst these correlated events, the relationship to hnRNP H insolubility varied; for example, splicing changes at *hnRNPA3* followed a linear relationship similar to *BBS*, *GABRG2* and *CCDC136*; in turn, splicing changes at *hnRNPH1* occurred only when hnRNP H insolubility was above 60%, primarily in FTD patients and some C9 patients. *RIOK3* and several of the top 100 most correlated events with hnRNP H insolubility presented exponential trends similar to *hnRNPH1* splicing defects. Perhaps this indicates that some events are regulated in compensatory fashions by many factors (see below), and only undergo change in inclusion when soluble levels of multiple RBPs are severely compromised.

Broadly, the 100 events with the highest correlation with hnRNP H insolubility displayed variable splicing strength (intron exclusion ratios) across the cohort (*Figure 6b*; events listed in *Figure 6—figure supplement 1*), analogous to the pattern seen for all events (*Figure 5e*). Notably, among the highly correlated DS events were many in transcripts encoding RBPs (*Figure 6c*), such as splicing regulators, which generally followed the same pattern of PSI changes relative to insolubility in each group (*Figure 6d*; heatmap events listed in *Figure 6—figure supplement 2*; all events listed in *Supplementary file 1D*). Interestingly, several of the events found to affect RBPs (FUS exon 7, hnRNP A1 exon 8, hnRNP D exon 7 and TIA-1 exon 5) have been the subject of intensive inquiry for their potential roles in disease (*Deshaies et al., 2018*; *Zhou et al., 2013*) as well regulating splicing and interactions with other hnRNPs (*Gueroussov et al., 2017*; *Izquierdo and Valcárcel, 2007*). This observation raises the possibility that insolubility of the disease-associated RBPs leads to specific alterations in RNA splicing that compound defects in those same RBPs and others, thereby exacerbating the magnitude and extent of splicing dysregulation (see Discussion).

Beyond the RBPs, several of the DS events were consistent with possible direct roles in disease pathology. One especially noteworthy example was the increased skipping of MAPT exon 10, observed in C9, ALShigh and FTD (*Supplementary file 1B*). Alternative inclusion of this exon perturbs a delicate balance between 3-repeat and 4-repeat Tau, and genetic mutations in exon 10 or the surrounding intron are a significant cause of familial FTD (*Ghetti et al., 2015*). While it is compelling to think that specific splicing changes could drive disease pathologies, the myriad of biologically interesting events suggests caution in attributing too much meaning to any individual event.

We next recapitulated this stratification of patient groups using hierarchical clustering. Using the 489 differential events that were common to all groups (488 common to ALShigh, C9 and FTD, and one common to ALSlow as well) (*Figure 6—figure supplement 3*), patients with similar insolubility measurements were grouped close to one another. To further support the hypothesis that all DS events are due to insolubility and not particular clinical distinctions, clustering of patients based on the top events that were unique to each grouping (20 from each ALShigh, C9 and FTD, and all three events in ALSlow) also ordered the patients according to insolubility (*Figure 6—figure supplement 4*). Regardless of whether clustering was performed with either the most shared (*Figure 6—figure supplement 3*) or the most divergent (*Figure 6—figure supplement 4*) events, the two groups separated by the farthest distance contained almost exactly the same patients, whereas the controls (neurological normal and SOD1) clustered together with the patients with the lowest insolubility,

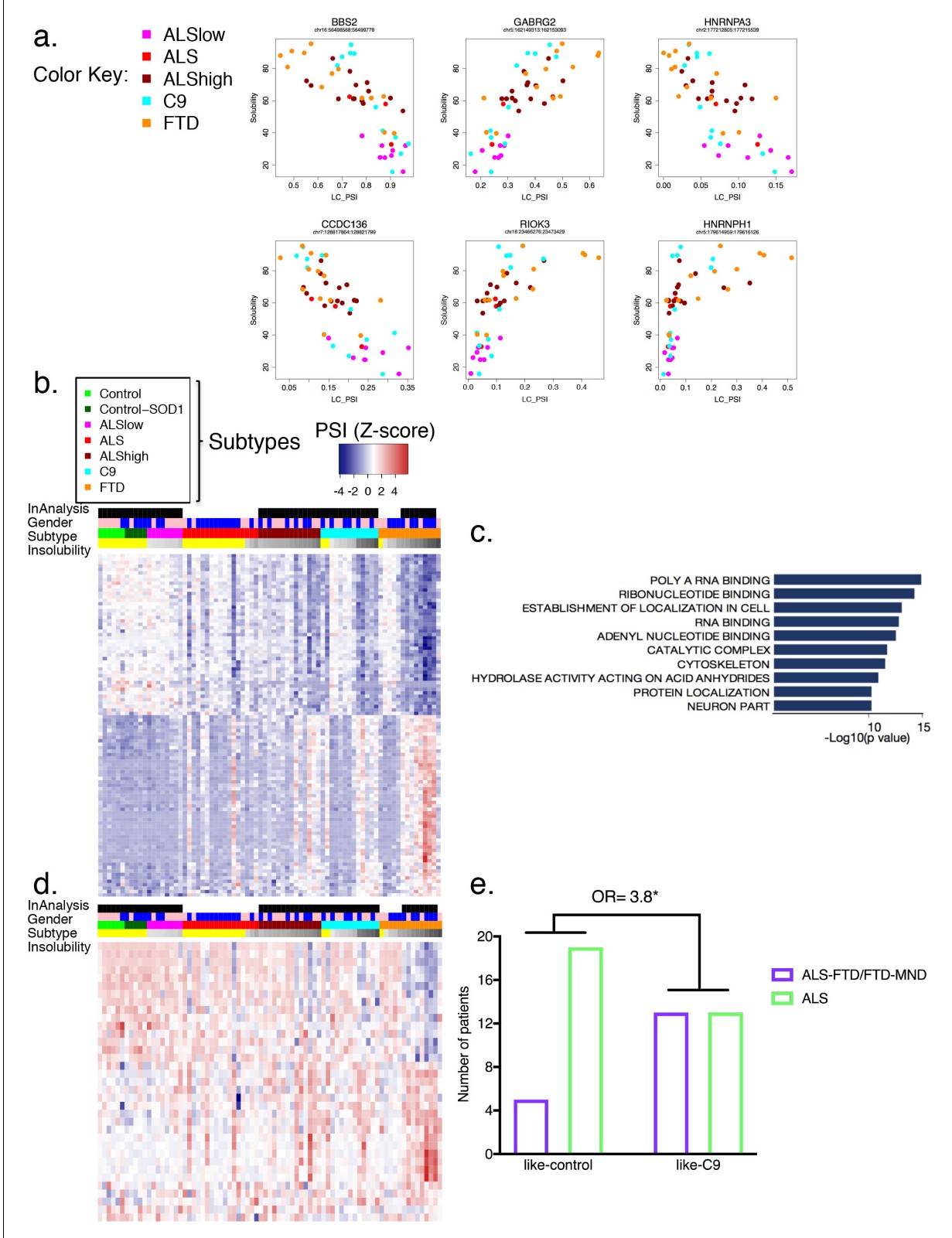

**Figure 6.** Global splicing differences correlate to hnRNP H insolubility. (a) Examples of events amongst the 100-most correlated to insolubility, with Leafcutter (LC) PSI plotted on the X-axis and insolubility (%) on the Y-axis. DS events are shown for the following genes (clockwise, from upper left): *BBS2*, *GABRG2*, *HNRNPA3*, *HNRNPH1*, *RIOK3* and *CCDC136* (events listed in **Supplementary file 1C**). (b) Heatmap of 100 events most correlated/anti-correlated (absolute value Spearman coefficient) with hnRNP H insolubility. Blue-red color scale represents Z-score for normalized PSI. Four rows above

*Figure 6 continued on next page*

*Figure 6 continued*

heatmap display whether each sample was included in the groupings used for the identification of DS events, termed 'In analysis' (black = yes, white = no), gender (pink = female, blue = male), Patient subtype (see key), and percent insolubility hnRNP H (180,000 x G), ranging from light (low insolubility) to dark (high insolubility) gray. Yellow is shown for cases without insolubility data. Patients are arranged by subtype, with increasing insolubility from left to right, where data is available. (c) Gene set enrichment analysis of top 200 dysregulated splicing events correlated to hnRNP H insolubility. (d) Heatmap of significant DS events that occur in a set of common splicing factor genes, simplified to the single most changed event per gene (hnRNPs, SR proteins, ALS-related RBPs or SMN; events listed in *Supplementary file 1D*). The same four rows apply as in (b). (e) Bar graph depicting that patients like-C9 are 3.8-times more likely to have symptoms of FTD than patients like-control (Fisher's exact test, Odds ratio (OR) = 3.8, p=0.032).

DOI: https://doi.org/10.7554/eLife.37754.012

The following figure supplements are available for figure 6:

**Figure supplement 1.** Events in *Figure 6b* heatmap as they appear in order.
DOI: https://doi.org/10.7554/eLife.37754.013
**Figure supplement 2.** Events in *Figure 6d* heatmap as they appear in order.
DOI: https://doi.org/10.7554/eLife.37754.014
**Figure supplement 3.** Heatmap of intersecting events.
DOI: https://doi.org/10.7554/eLife.37754.015
**Figure supplement 4.** Heatmap of subtype-specific DS events.
DOI: https://doi.org/10.7554/eLife.37754.016
**Figure supplement 5.** A model for a multi-RBP proteinopathy and splicing dysregulation as a common mechanism underling ALS/FTD pathogenesis.
DOI: https://doi.org/10.7554/eLife.37754.017

including samples not in the original ALSlow group. In summary, we conclude that widespread, global splicing differences occur in ALS and FTD, and these can be explained by biochemical insolubility of RBPs, with corresponding severity across distinct brain regions.

## Like-C9 patients have a higher incidence of FTD

In the above analysis based on solubility groupings, the FTD samples were noteworthy in that they displayed the highest hnRNP H insolubility, as well as the most numerous and acute PSI changes. However, several sporadic and C9 ALS patients with high insolubility had similar DS patterns, implying that the excessive events were not somehow specific to the FTD clinical diagnosis, but rather the result of exceptionally high insolubility of RBPs, which occurs at a greater rate in FTD than ALS. This suggests a possible relationship between patients identified as like-C9 and FTD.

To determine if like C9-patients were indeed more likely to be diagnosed with FTD, we returned to our initial assignments of like-C9 and like-control. Of the 50 patients in this analysis, 18 had symptoms of FTD, and were clinically diagnosed with either ALS-FTD or pure FTD, while the remaining 32 patients were diagnosed with ALS without cognitive or behavioral impairment. The majority (13/18; 72%) of the patients characterized by symptoms of FTD were like-C9, while only five of the patients with FTD symptoms were like-control. We compared these ratios using a Fisher's exact test, and found that like-C9 patients were on average 3.8 times more likely to have symptoms of FTD versus like-control patients (*Figure 6e*, p=0.032). Strikingly, these findings mirror what has been reported for C9 patients, who are approximately four times more likely than the ALS population at large to receive a diagnosis of FTD (*Byrne et al., 2012*). Thus we conclude that this distinction of like-C9 based upon splicing and accompanying biochemical insolubility confers clinical susceptibility to FTD, just as occurs in authentic C9ALS patients.

## Discussion

In this study, we have characterized a diverse cohort of ALS/FTD patient brains and identified a signature of graded splicing dysregulation that correlates in magnitude with insolubility of several abundant and disease-relevant RBPs. Our work provides evidence that aberrant alternative splicing is a common molecular feature of C9ALS/FTD and sporadic ALS/FTD patients, and importantly, that these changes reflect underlying biochemical changes that manifest in RBP aggregation. A unified vision of what causes ALS and FTD at the molecular levels has remained highly speculative despite great advances in our understanding of the genetics that contribute to this disease spectrum

(*Taylor et al., 2016*). Thus, our work is significant in that it provides a compelling case for the existence of a shared biochemical mechanism that occurs in ALS, FTD and ALS/FTD, and which can be rationalized in the context of other known disease mutations and pathologies.

RNA splicing defects have previously been reported in C9ALS (*Conlon et al., 2016*) and sporadic ALS/FTD (*Ling et al., 2015*; *Prudencio et al., 2015*). However, in the absence of mechanistic insights into the origins of these defects in sporadic patients, the assumed culprit has been TDP-43 deficiency, reflecting the almost universal observation of TDP-43 histopathology (*Polymenidou et al., 2011*; *Lagier-Tourenne et al., 2012*; *Ling et al., 2015*). Through stratification of patient samples based on biochemical insolubility of hnRNP H, and by extension TDP-43, we found that TDP-43 pathology is not sufficient to confer detectable splicing differences relative to neurologically normal controls or SOD1 ALS patients. Thus, our data suggest a model (*Figure 6— figure supplement 5*) in which TDP-43 is just one of several affected RBPs. Our finding that biochemical insolubility of TDP-43 correlates with splicing defects as well as with insolubility of several other RBPs casts doubt on the TDP-43-centric view of ALS (*Lagier-Tourenne and Cleveland, 2009*; *Scotter et al., 2015*). Given the decoupling of visible pathology from biochemical insolubility, with only the latter correlated with splicing dysregulation, our results strongly suggest that ALS/FTD should be considered a multiRBP proteinopathy rather than a TDP-43 proteinopathy.

The fact that approximately half of sporadic ALS/FTD patient brains do not display significant splicing defects or insoluble protein seemingly contradicts our assertion that this is the common mechanism underlying disease. One possibility is that an entirely distinct mechanism drives neurodegeneration in these patients, perhaps one similar to SOD1ALS in which soluble hnRNP H and splicing profiles resemble neurologically normal controls. Alternatively, the range of abnormality in sALS patients we observed may reflect what we detected in C9ALS/FTD (*Conlon et al., 2016*). Specifically, some C9 patients displayed severe splicing changes, while others were indistinguishable from normal controls. One possible explanation for this is that in both C9s and sporadic cases, some patients display these changes only on a limited scale. For example, splicing and insolubility changes may be confined to only the most vulnerable cell types in patients (sporadic and C9) that were modestly different from control. As a result, changes in splicing or solubility that occur in a minority subpopulation of cells could be diluted in bulk studies by more numerous transcripts derived from unaffected cell types. A future goal will be to try to investigate whether such specific splicing events indeed exist that are unique or disproportionately relevant to affected neuronal populations.

Another interpretation of our data is that detectable splicing changes and highly insoluble protein levels may only become evident in the later stages of a progressive coalescence of RBPs (*Lin et al., 2015*; *Murakami et al., 2015*; *Patel et al., 2015*). In such a model, splicing changes, while undoubtedly contributory, may not be the sole, or critical, event driving disease onset or progression, but rather a consequence of perturbations in the equilibrium of soluble versus aggregated RBPs (*Figure 6—figure supplement 5*). Many studies of ALS/FTD disease mechanisms have focused on a complex equilibrium between differentially concentrated RBPs in physically distinct phases. Importantly, many factors, including mutations known to cause disease (*Kim et al., 2013*; *Murakami et al., 2015*; *Patel et al., 2015*; *Martinez et al., 2016*; *Fay et al., 2017*; *Mackenzie et al., 2017*), are capable of influencing the balance between protein concentrations in each phase over time (*Ramaswami et al., 2013*). Our model (*Figure 6—figure supplement 5*) proposes that all ALS/FTD patients fall at varying points beyond a hypothetical 'toxicity boundary' that would be analogous to the phase boundaries that describe physiological granule formation (*Weber and Brangwynne, 2012*; *Elbaum-Garfinkle et al., 2015*), only unidirectional. The magnitude of this imbalance, or position past a theoretical tipping point, would thus be related to increased abundance of biochemically insoluble protein and decreased levels of functional RBPs, leading to aberrant splicing.

The above model depicts RBP insolubility as lying upstream of splicing defects. However, it is worth considering the possibility that splicing changes lead to, or at least contribute to, insolubility. Notably, two events found to affect the RBPs hnRNP D and hnRNP A1 (exons 7 and 8, respectively) have been shown to cause inclusion of glycine-tyrosine (GY)-rich intrinsically disordered regions (IDRs) in their respective proteins (*Gueroussov et al., 2017*). In our analysis, we observed increased inclusion of hnRNP D exon 7, leading to an isoform with an extended GY-rich region that makes increased contacts with other hnRNPs, thereby potentially promoting overly stable RBP complexes. Inclusion of hnRNP A1 exon 8 also extends its IDR, although the difference in the number of GY-

repeats is not as strong as for hnRNP D (*Gueroussov et al., 2017*). Somewhat surprisingly, hnRNP A1 exon 8 was found in our data to be more skipped as RBP insolubility increased, which seems to contradict a recent study suggesting that reduced levels of TDP-43 promote inclusion of this exon (*Deshaies et al., 2018*). In contrast, the loss of soluble FUS, which regulates its own transcript, may explain FUS splicing defects in our data. Normally, FUS protein represses inclusion of exon 7, leading to a FUS isoform that undergoes nonsense-mediated decay in a self-regulating loop (*Zhou et al., 2013*). Thus, it may be that reduced levels of soluble FUS lead to the observed increase in exon 7 inclusion, thereby resulting in excess levels of FUS that either seed RBP aggregates or 'pile on' to existing ones. Such a mechanism, whereby splicing changes affecting one or a few RBPs act as 'drivers' of RBP aggregation, can also be envisioned for other events in our analysis, for example, hnRNP H1, which we found undergoes differential splicing in its gly-rich IRD. Regardless of whether insolubility is upstream of splicing changes, or splicing changes precipitate insolubility, these events, and likely others, highlight how the two processes may reinforce one another in a vicious cycle, ultimately leading to the severe insolubility/splicing dysregulation we have documented in the like-C9 ALS and FTD cases.

If all ALS-FTD indeed converges on the same mechanism then what factors dictate the onset and progression of disease at the clinical level? This is not straightforward, even when a known mutation underlies disease. In the case of C9, understanding how a single genetic polymorphism can lead to two different disorders, or a combination of the two, is perplexing. Further complicating matters, the C9 expansion has been associated with scarce disease incidence in a wide spectrum of neurodegenerative and psychiatric conditions, including Huntington's Disease, Parkinson's Disease, and Schizophrenia (*Hensman Moss et al., 2014*; *Lesage et al., 2013*; *McLaughlin et al., 2017*). It is unclear if these different patterns of neuronal degeneration reflect intra-patient variation in toxic burden across cell types/brain regions (*Mackenzie et al., 2014*), or if particular cell-types have inherently lower tolerance to toxic threats that are uniform across an individual (*Nijssen et al., 2017*). It may be that a combination of burden mosaicism and varying toxicity thresholds of distinct cell types converge to affect disease progression, such that mutation alone cannot predict the disease course of a given individual (*Van Mossevelde et al., 2017*). In our C9 model, these hypothetical burdens may be manifested and/or enhanced by transcriptional activity of the expanded locus (*Liu et al., 2014*), heightened ratio of aggregated to soluble hnRNP H, and increasingly severe splicing changes. For sALS/FTD, we know the latter two occur, but as of yet, what triggers this remains a mystery.

The variation in disease severity implied in our model also suggests a natural progression of disease from ALS to FTD along this clinical spectrum. Our results suggest that FTD patients are further past this hypothetical boundary than are patients with pure ALS, as reflected by the highest values of RBP insolubility, and most global splicing changes. Given that all of the patients in our cohort with FTD also displayed MN loss, but not all ALS patients displayed signs of frontal cortex deterioration, we propose that the distinct neuronal populations affected by this disease spectrum have differential vulnerabilities to the alterations in granule dynamics that define the toxicity boundary. The heightened sensitivity of spinal MNs implied by this gradient parallels what is known about SMA, a disease where subtle alterations in the concentration of the protein SMN cause highly specific degeneration of spinal MNs, with further increasing deficiency in SMN resulting in wider systemic dysfunction (*Tu et al., 2017*). In both cases, disease mechanisms have been suggested to converge on perturbations of granule dynamics (*Zhang et al., 2006*; *Shan et al., 2010*; *Ishihara et al., 2013*) and RNA splicing (*Tsuiji et al., 2013*; *Zhang et al., 2008*).

The observation that defects in mRNA splicing may have relevance to ALS was made 20 years ago, through examination of the disease-relevant *EAAT2* gene product (*Lin et al., 1998*). In retrospect, our findings shed new light on the observation that splicing changes could be found region-specifically, and in most, but not all, sporadic patients. In addition to providing a potential mechanistic basis for this observation, our data can fit it into a broader mechanistic context that correlates changes in mRNA splicing with changes in RBP solubility. While the ubiquity of TDP-43 pathology to ALS/FTD, and the protein's known roles in RNA processing, brought RNA biology to the forefront of research on these diseases, we now conclude that RNA processing defects are not simply correlated to visible pathology. Nonetheless, the microscopically invisible fluctuations in RBP concentration that we have described biochemically imply that this complex spectrum of disease states can be reached through many paths, some genetically mediated, some potentially driven by environment and

experience. Our discovery that a large fraction of all ALS/FTD is characterized by a 'like-C9' signature and reflects a 'multiRBP proteinopathy' thus provides a unifying theme for these complex and lethal diseases.

# Materials and methods

**Key resources table**

| Reagent type (species) or resource | Designation | Source or reference | Identifiers | Additional information |
|---|---|---|---|---|
| Biological sample (human) | Patient brains samples | NA | All identifiers are provided in *Supplementary file 1A* | |
| Antibody | hnRNP H (rabbit) | Bethyl, A300-511 | | |
| Antibody | hnRNP H (rabbit, Co-IP) | ThermoFisher, PA5-27610 | | |
| Antibody | TDP-43 (rabbit) | Proteintech, 10782–2-AP | | |
| Antibody | TDP-43 human specific (mouse) | Proteintech, 60019–2-Ig | | |
| Antibody | hnRNP A1 (mouse) | Sigma, 4B10 | | |
| Antibody | FUS (mouse) | Santa Cruz Biotechnologies, H-6 | | |
| Antibody | GAPDH (rabbit) | Sigma, G9545 | | |
| Antibody | U2AF65 (mouse) | Sigma, U4758 | | |
| Antibody | SMN (mouse) | Sigma, S2944 | | |
| Antibody | Histone H3 (rabbit) | Abcam, 1791 | | |
| Antibody | FLAG-M2 (mouse) | Sigma, F1804 | | |
| Antibody | Actin (rabbit) | Sigma, A2066 | | |
| Antibody | C9ORF72 (rabbit) | Novus, 1086CGP | | |
| Software, algorithm | Leafcutter | | *Li et al. (2018)* | |

## Acquisition of patient materials

Human patient brains were donated for research by next of kin. Samples were acquired from the New York Brain Bank. C9 expansion testing was performed by repeat-primed PCR in the Clinical Pathology lab of Columbia University Medical Center. All available clinical and pathological records were collected and used to summarize patient demographics and disease features. Additional samples were obtained by the New York Genome Center as part of the Target ALS post-mortem core.

## RT-PCR

Total RNA was extracted from patient cerebellum, polyA selected, reverse-transcribed, and amplified by $^{32}$P-PCR for 33 cycles with established primers as previously described (*Conlon et al., 2016*). Products were electrophoresed side-by-side on 6% native PAGE and exposed to phosphor. Percent inclusions were measured using ImageQuant, and replicate data from two independent RNA extractions per sample were graphed and analyzed with Graphpad for Prism. All primer sequences for hnRNP H (*Conlon et al., 2016*) and TDP-43 (*Tollervey et al., 2011*) targets were previously published. Primer sequences (5′−3′) for ACHE are as follows: 4F: GAACCGCTTCCTCCCCAAATT; 5aR: CAGCCTCCCCATGGGTGAA; 5bR: GTGGAACTCGGCCTTCCACT.

## Biochemical fractionation

Biochemical fractionation using Sarkosyl-containing buffers was performed as previously described (*Conlon et al., 2016*), with 21,000 x g centrifugation (15 minutes) substituted for 180,000 x g (30 minutes) where indicated. All three fractions, soluble, sarkosyl soluble and sarkosyl insoluble, were western blotted simultaneously, and the band intensities of each fraction were quantified using ImageJ from a single exposure.

## Western blotting

Western blotting was performed as previously described (*Conlon et al., 2016*). Antibodies used in include Actin (Sigma, A2066), hnRNP H (Bethyl, A300-511), TDP-43 (Proteintech, 10782–2-AP), FUS (Santa Cruz Biotechnologies, H-6), GAPDH (Sigma, G9545), hnRNP A1 (Sigma, 4B10), Histone H3 (Abcam, 1791), SMN (Sigma, S2944), U2AF65 (Sigma, U4758), C9ORF72 (Novus, 1086CGP), FLAG-M2 (Sigma, F1804). All primary antibodies were incubated in Pierce Protein-free blocking buffer.

## Co-Immunoprecipitation

Fractionation was performed as described (*Conlon et al., 2016*) with the following exceptions: after the 180,000 x g centrifugation, the insoluble pellet was washed briefly with milliQ water, then incubated in 20 mM Tris, 100 mM KCl (supplemented with 10 mM phenylmethylsulfonyl fluoride (PMSF) and 10 µg/ml each of aprotinin, leupeptin and pepstatin [a/l/p]) equal to 5% the volume of the Sarkosysl soluble-fraction. The pellet was incubated with rotation for 24 hr at 4°C, then centrifuged briefly at 5000 x g. One microliter of elutant (without debris) was diluted into a final volume of 50 µl IP buffer (50 mM Tris pH 7.5, 150 mM NaCl, 0.5% NP-40, 10% glycerol with 10 mM PMSF and 10 µg/ml each of a/l/p) with either 0.5 ug TDP-43 antibody (Proteintech, 60019–2-Ig), 0.5 µg FLAG-M2 antibody (Sigma) or nothing, and incubated at 4°C for 60–100 min with rotation. One ul of input was removed and stored at 4°C. Protein A/G beads (Pierce; 10 µl per sample) were washed three times with 1 mL IP buffer, the resuspended in 25 µl IP buffer. Beads were then mixed with the 50 µl sample, and incubated for 16–24 hr at 4°C with rotation. Supernatants were separated from beads on a magnetic stand, and beads were washed three times with 1 mL high-salt IP buffer (50 mM Tris pH 7.5, 500 mM NaCl, 0.5% NP-40, 10% glycerol with 10 mM PMSF and 10 µg/ml each of a/l/p), with 5 min rotation at 4°C with each wash. Beads were centrifuged briefly (<10 s) at 500 x g after the third wash, and residual liquid was fully removed. Beads were eluted for 5–8 min with 100 mM glycine, pH 2.0 at room temperature, then immediately neutralized with 100 mM Tris pH 10.0. The eluent was mixed with 4X SDS sample buffer, boiled, electrophoresed by 8% SDS-PAGE, and western blotted with hnRNP H (ThermoFisher, PA5-27610) and TDP-43 (Proteintech, 10782–2-AP) antibodies.

## RNA-sequencing

Total RNA was extracted from flash-frozen post-mortem cerebellum tissue in Trizol/Chloroform and purified using a Qiagen RNeasy minikit column. Starting form 500 ng total RNA input, rRNA depletion and library preparation were performed according to the manufacturer's conditions using the KAPA Stranded RNA-Seq Kit with RiboErase and unique Illumina-compatible indexes (NEXTflex RNA-seq Barcodes, BioScientific). Multiplexed libraries 550 bp in length were sequenced PE 125 (Illumina HiSeq 2500).

## Computational methods

Forty to sixty million filtered reads were aligned to GRCh38 using STAR (2.5.2a) (*Dobin and Gingeras, 2015*). Differential splicing analysis was performed using the LeafCutter annotation-free algorithm (*Li et al., 2018*). The package was implemented with default parameter settings per the author's recommendations: at least 50 split reads must support the cluster with introns up to 500 kb. Only clusters with p-adjusted <0.1 were considered for further analysis (a lower p-adjusted cutoff proved to be too stringent, with no significant clusters listing for the ALSlow analysis). We used gender as an additional co-variate to prevent events attributable to gender from cluttering the results. Variations in intron splicing were considered significant in ALS/FTD when the Percentage Spliced Index (PSI) compared to control differed by more than 10% ($|\Delta PSI| \geq 0.1$) with 10% FDR cutoff. Differential splicing events were then mapped to gene names to identify events of interest using Gencode version 25.

## Acknowledgements

This work was supported by NIH grant R35 GM 118136 to JLM. EGC was supported in part by NIH training grant 5T32GM008798. RNA sequencing and related analyses at the NYGC were supported by the ALS Association (grant 15-LGCA-234) and the Tow Foundation, which also provided direct support for the Eleanor and Lou Gehrig ALS Center (NAS). We acknowledge the Target ALS

postmortem tissue core, as well as the New York Brain Bank for providing us with human brain samples. We thank Dr. Lawrence Honig of the Taub Institute/ADRC for providing FTD samples. We also thank Aarti Sharma and Beatriz Blanco-Redondo for many helpful discussions that helped shape this study.

## Additional information

### Group author details

**The New York Genome Center ALS Consortium**
**Hemali Phatnani**: Center for Genomics of Neurodegenerative Diseases, New York Genome Center, New York, United States; **Justin Kwan**: Department of Neurology, University of Maryland School of Medicine, University of Maryland ALS Clinic, Baltimore, United States; **Dhruv Sareen**: Cedars-Sinai Department of Biomedical Sciences, Board of Governors Regenerative Medicine Institute and Brain Program, Cedars-Sinai Medical Center, Los Angeles, United States; Department of Medicine, University of California, Los Angeles, United States; **James R Broach**: Department of Biochemistry and Molecular Biology, Penn State Institute for Personalized Medicine, The Pennsylvania State University, Hershey, United States; **Zachary Simmons**: Department of Neurology, The Pennsylvania State University, Hershey, United States; **Ximena Arcila-Londono**: Department of Neurology, Henry Ford Health System, Detroit, United States; **Edward B Lee**: Department of Pathology and Laboratory Medicine, Perelman School of Medicine, University of Pennsylvania, Philadelphia, United States; **Vivianna M Van Deerlin**: Department of Pathology and Laboratory Medicine, Perelman School of Medicine, University of Pennsylvania, Philadelphia, United States; **Neil A Shneider**: Department of Neurology, Center for Motor Neuron Biology and Disease, Institute for Genomic Medicine, Columbia University, New York, United States; **Ernest Fraenkel**: Department of Biological Engineering, Massachusetts Institute of Technology, Cambridge, United States; **Lyle W Ostrow**: Department of Neurology, Johns Hopkins School of Medicine, Baltimore, United States; **Frank Baas**: Department of Neurogenetics, Academic Medical Centre, Amsterdam and Leiden University Medical Center, Leiden, Netherlands; **Noah Zaitlen**: Department of Medicine, Lung Biology Center, University of California, San Francisco, San Francisco, United States; **James D Berry**: ALS Multidisciplinary Clinic, Neuromuscular Division, Department of Neurology, Harvard Medical School, Boston, United States; Neurological Clinical Research Institute, Massachusetts General Hospital, Boston, United States; **Andrea Malaspina**: Centre for Neuroscience and Trauma, Blizard Institute, Barts and The London School of Medicine and Dentistry, Queen Mary University of London, London, United Kingdom; Department of Neurology, Basildon University Hospital, Basildon, United Kingdom; **Pietro Fratta**: Institute of Neurology, National Hospital for Neurology and Neurosurgery, University College London, London, United Kingdom; **Gregory A Cox**: The Jackson Laboratory, Bar Harbor, United States; **Leslie M Thompson**: Department of Psychiatry and Human Behavior and Department of Biological Chemistry, School of Medicine, University of California, Irvine, Irvine, United States; Department of Neurobiology and Behavior, School of Biological Sciences, University of California, Irvine, Irvine, United States; **Steve Finkbeiner**: Taube/Koret Center for Neurodegenerative Disease Research, Roddenberry Center for Stem Cell Biology and Medicine, Gladstone Institute, San Francisco, United States; **Efthimios Dardiotis**: Department of Neurology and Sensory Organs, University of Thessaly, Thessaly, Greece; **Timothy M Miller**: Department of Neurology, Washington University in St. Louis, St. Louis, United States; **Siddharthan Chandran**: Centre for Clinical Brain Sciences, Anne Rowling Regenerative Neurology Clinic, Euan MacDonald Centre for Motor Neurone Disease Research, University of Edinburgh, Edinburgh, United Kingdom; **Suvankar Pal**: Centre for Clinical Brain Sciences, Anne Rowling Regenerative Neurology Clinic, Euan MacDonald Centre for Motor Neurone Disease Research, University of Edinburgh, Edinburgh, United Kingdom; **Eran Hornstein**: Department of Molecular Genetics, Weizmann Institute of Science, Rehovot, Israel; **Daniel J MacGowan**: Department of Neurology, Icahn School of Medicine at Mount Sinai, New York, United States; **Terry Heiman-Patterson**: Center for Neurodegenerative Disorders, Department of Neurology, the Lewis Katz School of Medicine, Temple University, Philadelphia, United States; **Molly G Hammell**: Cold Spring Harbor Laboratory, Cold Spring Harbor, United States; **Nikolaos A Patsopoulos**: Computer Science and Systems Biology Program, Ann

Romney Center for Neurological Diseases, Department of Neurology, Brigham and Women's Hospital, Harvard Medical School, Boston, United States; Division of Genetics in Department of Medicine, Brigham and Women's Hospital, Harvard Medical School, Boston, United States; Program in Medical and Population Genetics, Broad Institute, Cambridge, United States; **Joshua Dubnau**: Department of Anesthesiology, Stony Brook University, Stony Brook, United States; **Avindra Nath**: Section of Infections of the Nervous System, National Institute of Neurological Disorders and Stroke, NIH, Bethesda, United States

## Competing interests
James L Manley: Senior editor, *eLife*. The other authors declare that no competing interests exist.

## Funding

| Funder | Grant reference number | Author |
| --- | --- | --- |
| NIH Office of the Director | R35 GM 118136 | James L Manley |
| NIH Office of the Director | 5T32GM008798 | Erin G Conlon |
| Amyotrophic Lateral Sclerosis Association | 15-LGCA-234 | Hemali Phatnani |

The funders had no role in study design, data collection and interpretation, or the decision to submit the work for publication.

## Author contributions
Erin G Conlon, Conceptualization, Formal analysis, Investigation, Visualization, Methodology, Writing—original draft; Delphine Fagegaltier, Conceptualization, Data curation, Formal analysis, Visualization, Writing—review and editing; Phaedra Agius, Data curation, Formal analysis, Visualization, Writing—review and editing; Julia Davis-Porada, Investigation, Writing—review and editing; James Gregory, Methodology, Writing—review and editing; Isabel Hubbard, Kristy Kang, Investigation; Duyang Kim, Resources; The New York Genome Center ALS Consortium, Resources, Funding acquisition; Hemali Phatnani, Conceptualization, Funding acquisition, Methodology, Writing—review and editing; Neil A Shneider, Conceptualization, Resources, Supervision, Funding acquisition, Methodology, Project administration, Writing—review and editing; James L Manley, Conceptualization, Supervision, Funding acquisition, Methodology, Writing—original draft

## Author ORCIDs
Neil A Shneider ⓘ http://orcid.org/0000-0002-3223-7366
James L Manley ⓘ https://orcid.org/0000-0002-8341-1459

## Ethics
Human subjects: Human post-mortem brain samples were donated for research purposes by next of kin.

## Decision letter and Author response
Decision letter https://doi.org/10.7554/eLife.37754.024
Author response https://doi.org/10.7554/eLife.37754.025

# Additional files

## Supplementary files
• Supplementary file 1. (**A**) Patient clinical information. (**B**) All significant DS events with greater than 10% difference in PSI. All events found to be significantly different between each patient group (FTD, ALShigh, ALSlow, C9) and control (|ΔPSI| ≥ 0.1). Events are arranged from greatest to smallest difference in ΔPSI, with ΔPSI values listed. Events without significant change in a given group are denoted with zeroes. All coordinates are mapped to GRCh38/hg38. (**C**) Significant DS events most

correlated to hnRNP H insolubility. Events and spearman rank correlation coefficients are provided. All coordinates are mapped to GRCh38/hg38. (**D**) Significant DS events occurring in common RBP genes. All events occurring in a set of common splicing factor genes (hnRNPs, SR proteins, ALS-related RBPs or SMN) amongst the events found to be significantly different between each patient group (FTD, ALShigh, ALSlow, C9) and control (no PSI filter). Events without significant change in a given group are denoted with zeroes. All coordinates are mapped to GRCh38/hg38. (**E**) NYGC ALS Consortium Members.

DOI: https://doi.org/10.7554/eLife.37754.018

• Transparent reporting form

DOI: https://doi.org/10.7554/eLife.37754.019

### Data availability

All raw RNASeq data from Target ALS samples are immediately made publicly available. Access to the data can be requested by emailing ALSData@nygenome.org. All RNASeq data from the ALS Consortium are made immediately available to all members of the Consortium and with other Consortia with whom we have a reciprocal sharing arrangement. Data have been deposited in the National Center for Biotechnology Information Gene Expression Omnibus (GEO) (accession no. GSE116622).

The following dataset was generated:

| Author(s) | Year | Dataset title | Dataset URL | Database, license, and accessibility information |
|---|---|---|---|---|
| James L Manley | 2018 | Data from: Unexpected similarities between C9ORF72 and sporadic forms of ALS/FTD suggest a common disease mechanism | http://www.ncbi.nlm.nih.gov/geo/query/acc.cgi?acc=GSE116622 | Publicly available at the NCBI Gene Expression Omnibus (accession no: GSE116622) |

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
