## [Decision Letter]

Thank you for submitting your article "Unexpected similarities between C9ORF72 and sporadic forms of ALS/FTD suggest a common disease mechanism" for consideration by *eLife*. Your article has been reviewed by three peer reviewers, and the evaluation has been overseen by Douglas Black as the Reviewing Editor, and Kevin Struhl as the Senior Editor The following individuals involved in review of your submission have agreed to reveal their identity: Mani Ramaswami (Reviewer #1); Aaron D Gitler (Reviewer #2).

The reviewers have discussed the reviews with one another and the Reviewing Editor has drafted this decision to help you prepare a revised submission.

Summary:

This study from Conlon et al, is submitted as an advance that extends their previous *eLife* paper. In the earlier study, the authors examined the GGGGCC hexanucleotide repeat expansion in the *C9ORF72* gene that causes ALS, and found that the expanded GGGGCC RNA bound to the splicing regulator hnRNP H. They further showed that the C9 mutation was associated with increased portions of the cellular hnRNP H being found in insoluble aggregates, and that C9 patient brains exhibited changes in splicing consistent with loss of hnRNP H function. ALS is a heterogeneous disease with multiple known causal mutations and with sporadic forms of unknown origin being highly prevalent. ALS is also often found in conjunction with frontal temporal dementia (FTD) and is thought to share a common origin with this and possibly other neurodegenerative disorders. In the new studies, the authors examined autopsied brains from sporadic non-C9 cases of ALS and FTD. Testing a panel of hnRNP H target exons, they identified a subset of cases of sporadic ALS (sALS) and particularly FTD with motor neuron degeneration that exhibit the patterns of splicing of C9 cerebellum. In this subset of sALS patients multiple hnRNP H enhanced exons showed lower inclusion and a few hnRNP H repressed exons showed higher levels of splicing, compared to other sALS patient brains or to control brains. Defining the sALS patients as "like-C9" in their splicing, or "like-control", they examined hnRNP H solubility in the cortices of these two patient groups. They found that in the like-C9 cortex a substantial portion of the hnRNP H was in the sarkosyl-insoluble pellet, and this was not seen in the like-control patient brains. The fraction of hnRNP H found in the sarkosyl pellet correlated strongly with the averaged splicing changes of the H dependent exons. In novel experiments examining the sarkosyl pellet fraction, they found that other RNA binding proteins known to be mutated or otherwise implicated in motor neuron disease also showed greater aggregation in the like-C9 patients compared to the like-control. These included TDP-43, FUS and hnRNP A1, whereas other proteins such as GAPDH were not aggregating. These results were extended by testing more selective pelleting assays and additional proteins to yield similar conclusions that the insolubility of a range of RNA binding proteins seen in ALS brain correlated with the observed splicing changes. The finding of multiple RBP's aggregating in parallel with hnRNP H led to a broader RNAseq assessment of splicing in the cerebelli of a range of ALS/FTD patients. Examining all the exon changes without regard to hnRNP H regulation, the authors found distinct regulatory clusters of splicing events in ALS patients that correlate with RBP aggregation, regardless of disease classification.

These data have several important implications for neurodegenerative disease. They define a subset of sporadic ALS patients with similar features to patients with the C9ORF72 mutation. The protein aggregation seen in C9 and like-C9 patients extends beyond hnRNP H to other RNA binding proteins mutated in other forms of ALS. In an intriguing observation, SMN1, whose loss of function is the cause of another motor neuron disease Spinal Muscular Atrophy, is also seen to aggregate in the like-C9 brains. Also interesting is the differential splicing of many RBP transcripts that the authors speculate could play an additional role in altering the splicing phenotype of ALS brain. Formation of TDP-43 inclusions in the motor cortex and elsewhere has long been used as a histopathological marker of ALS. However, the sarkosyl-insoluble RBP aggregates studied here as markers of like-C9 ALS (and that include TDP-43), were not seen to correlate with the histological TDP-43 inclusions seen across all ALS. These inclusions are speculated to perhaps be late events related to cell death rather than a causal factor in the disease. Direct experimental support for similarities between sporadic and genetic forms of ALS/FTD is important and valuable, and the description of ALS/FTD as RNA-binding proteinopathy instead of only a TDP-43 proteinopathy is an important new concept for the field. The reviewers all found this study to be of broad interest for both neurodegenerative disease and RNA biology, and to be suitable for publication as an *eLife* Research Advance. Major concerns were largely related to how the data are presented and discussed.

Essential revisions:

1) Throughout the paper and the previous study, the RNAseq analyses of splicing were carried out on cerebellar RNA, while the biochemical analyses of RNA binding protein aggregation used samples from the motor cortex, where the degeneration occurs. The authors cite valid practical reasons for doing the sequencing on cerebellar samples. However, they do not discuss possible implications of carrying out the two analyses on two very different brain regions. They should discuss whether the cerebellum also exhibits degeneration or is also forming the RBP aggregates. If not, they should discuss whether the splicing changes are truly a direct result of the protein aggregation. If the aggregates are formed across all brain regions, is there any cell type specificity to their presence? Are they found in other cells and tissues – similar to the loss of Smn1 in SMA? If they are found across all tissues, it is not clear what model the authors propose for the role of the protein aggregates. Different neurons and neuronal structures exhibit many differences in their gene expression and splicing and there will likely be cerebellar-specific and cortex-specific splicing events that are dependent on the aggregating proteins. If the disease targets motor neurons, shouldn't there be an assessment of the splicing changes in the tissue containing these cells?

2) The authors should also expand on the relationship of their new findings to the original splicing signature reported in 2016. Are the hnRNP H dependent splicing changes and the hnRNP H insolubility in C9 due to the expanded repeat RNAs sequestering hnRNP H? Or is it possible that the C9 aggregation is also caused other alterations in other RBPs that lead to the hnRNP H pathologies? A discussion of this will help readers assess the 2016 C9 data in light of their new findings.

3) Immunoblots for particular RNA binding proteins and other factors indicate that the formation of the aggregates is selective, but it is not clear how selective. Have the authors done shotgun mass spec identification of the proteins in the sarkosyl pellets? How many different proteins are aggregating? Are there other types of proteins in the aggregates besides RNA binders? If the MS data are available, it would potentially clarify several important questions regarding the aggregates and should be presented and discussed.

---

## [Author Response]

Essential revisions:1) Throughout the paper and the previous study, the RNAseq analyses of splicing were carried out on cerebellar RNA, while the biochemical analyses of RNA binding protein aggregation used samples from the motor cortex, where the degeneration occurs. The authors cite valid practical reasons for doing the sequencing on cerebellar samples. However, they do not discuss possible implications of carrying out the two analyses on two very different brain regions. They should discuss whether the cerebellum also exhibits degeneration or is also forming the RBP aggregates. If not, they should discuss whether the splicing changes are truly a direct result of the protein aggregation.

We agree with the reviewers that is an important issue. We have added the following sentences to the text at the end of the subsection “Like-C9 sALS/FTD display increased hnRNP H insolubility,” and provided new information (referencing “Purkinje Cell loss” status in Supplementary file 1) indicating that the cerebellum also exhibits degeneration.

“The correspondence between hnRNP H’s biochemical solubility and its predicted transcriptomic effects across distal brain regions from the same patients supports the notion that this is an intrinsic, brain-wide property of each patient. Reinforcing this view, we also examined all available post-mortem pathological reports, from 26 of the sporadic ALS/FTD patients. Significantly, 19 of these were noted to exhibit focal loss of Purkinje neurons in the cerebellum (Supplementary file 1), confirming that regions of the brain undergoing robustly measured splicing defects did indeed experience neuronal degeneration. This further justifies our choice to correlate changes measured in the cerebellum with changes in the cortex.”

If the aggregates are formed across all brain regions, is there any cell type specificity to their presence?

This is an interesting question, but in terms of a biochemical analysis (not immunofluorescence) would be virtually impossible to test given currently available methods.

Are they found in other cells and tissues – similar to the loss of Smn1 in SMA? If they are found across all tissues, it is not clear what model the authors propose for the role of the protein aggregates. Different neurons and neuronal structures exhibit many differences in their gene expression and splicing and there will likely be cerebellar-specific and cortex-specific splicing events that are dependent on the aggregating proteins.

This point reflects another facet of the brain’s complexity that we cannot account for properly. We agree that there may be important disease-relevant contributions of splicing events that are specific to particular neuronal subtypes. However, identifying those changes would require a sophisticated analysis of multiple brain regions, or ideally, cell types, from the same patients, which would be further complicated by the comparison of degenerated cell populations in patients to non-degenerated ones in controls. Although we think such an analysis represents an important future direction, it is not something that could be addressed in the current study. However, to acknowledge this important point, we have added the following sentences to the Discussion section.

“One possible explanation for this is that in both C9s and sporadic cases, some patients display these changes only on a limited scale. For example, splicing and insolubility changes may be confined to only the most vulnerable cell types in patients (sporadic and C9) that were modestly different from control. As a result, changes in splicing or solubility that occur in a minority subpopulation of cells could be diluted in bulk studies by more numerous transcripts derived from unaffected cell types. A future goal will be to try to investigate whether such specific splicing events indeed exist that are unique or disproportionately relevant to affected neuronal populations.”

If the disease targets motor neurons, shouldn't there be an assessment of the splicing changes in the tissue containing these cells?

We agree that such an analysis would be worthwhile. However, technical difficulties, such as lower yield of RNA and spatial heterogeneity in the distribution of cell types over large volumes of tissue, make this challenging. In any case, the strong concordance between the extent of RBP aggregation in motor cortex and the magnitude of missplicing in cerebellum make a very strong case that the splicing defects we observed occur throughout the brain. This is now stressed in the text. An interesting question though is whether there may be a subset of transcripts that are misspliced in specific cell types such as motor neurons. However, an analysis of the depth required to detect this would be well beyond the scope of this study, technical difficulties aside.

2) The authors should also expand on the relationship of their new findings to the original splicing signature reported in 2016. Are the hnRNP H dependent splicing changes and the hnRNP H insolubility in C9 due to the expanded repeat RNAs sequestering hnRNP H? Or is it possible that the C9 aggregation is also caused other alterations in other RBPs that lead to the hnRNP H pathologies? A discussion of this will help readers assess the 2016 C9 data in light of their new findings.

It is our view that there are multiple ‘seeds’ that can ultimately lead to similar aggregation, and that C9 RNA is probably a particularly strong ‘seed’. Examples of other potential ‘seeds’ are given in Figure 6—figure supplement 5. These include *TARDBP* mutation, *hnRNPA1* mutation, *FUS* mutation, stress or injury. We mention in the Discussion section that “many factors, including mutations known to cause disease […] are capable of influencing the balance between protein concentrations in each phase over time.” Ultimately, we do not know what initiates the aggregation described in sporadic patients in this study, or whether there is one common ‘seed’ amongst these patients. All we can say is that these changes are readily detectable in a large fraction of patients. Furthermore, we have discussed how aberrant splicing could itself be a ‘seed’ through the altered expression of RBP isoforms with differential tendencies to make protein-protein interactions that may be prone to aggregation (Discussion section).

Along these lines, we have also provided new experimental evidence (Figure 4l) that supports this ‘seed’ hypothesis. The high degree of correspondence between insoluble TDP-43 and insoluble hnRNP H can be explained by the fact that the two proteins co-aggregate in brain, as supported by co-immunoprecipitation from the sarkosyl insoluble pellet (Figure 4l). We have added the following to the text in subsection “Reciprocal insolubilities of hnRNP H and TDP-43 reveal TDP-43 LOF is exclusive to like-C9s”:

“One explanation for the similar degrees of hnRNP H and TDP-43 insolubility in the sporadic and C9 patient brains is that they coexist in the same physical aggregates. To test this hypothesis, we performed co-immunoprecipitation (co-IP) of hnRNP H from eluates of the sarkosyl insoluble pellets of two like-C9 brains using a TDP-43 antibody for IP (see Methods). Indeed, TDP-43 antibody successfully co-IPed hnRNP H, while a negative control antibody (FLAG-M2) or beads alone did not (Figure 4l). These results support a reciprocal relationship between insolubility of hnRNP H and TDP-43, whereby aggregates initially seeded by one protein can recruit the other, and vice versa.”

In the case of C9s, we previously provided evidence that the aggregates contain G-Q forming RNA, suggesting the expansion in C9ORF72 can seed aggregates of hnRNP H via direct protein-RNA interactions (Conlon et al., 2016). Our new results now suggest this can further seed aggregation of additional RBPs such as TDP-43. Thus, if the two proteins are capable of co-aggregating, we envision that insolubility of a factor such as TDP-43 could be ‘upstream’ of hnRNP H insolubility. However, this begs the question, what makes TDP-43 insoluble? We found no experimental evidence for direct sequestration of TDP-43 by the expanded RNA. Furthermore, we have shown in this study that existence of TDP-43 pathology in degenerating regions such as the motor cortex does not necessarily translate into biochemically insoluble TDP-43, thus refuting the potential argument that TDP-43 pathology could be the cause of the hnRNP H insolubility and splicing changes.

3) Immunoblots for particular RNA binding proteins and other factors indicate that the formation of the aggregates is selective, but it is not clear how selective. Have the authors done shotgun mass spec identification of the proteins in the sarkosyl pellets? How many different proteins are aggregating? Are there other types of proteins in the aggregates besides RNA binders? If the MS data are available, it would potentially clarify several important questions regarding the aggregates and should be presented and discussed.

We agree that such an analysis would be informative; however, we did not generate any mass spec data. Such an analysis would likely not be straightforward and quantitatively accurate, as eluting insoluble proteins from the sarkosyl-insoluble pellet would likely lead to biases towards detecting proteins that re-solubilized more readily. Nonetheless, the method of eluting the pellet in a Tris solution that we used to perform co-IP could perhaps be optimized to perform mass spec, and we believe this represents an interesting future direction.